# The role of tropopause polar vortices in the intensification of summer Arctic cyclones

Suzanne L. Gray[1], Kevin I. Hodges[1,2], Jonathan L. Vautrey[1,*], and John Methven[1]

[1]Department of Meteorology, University of Reading, Reading, RG6 6ET, UK
[2]National Centre for Atmospheric Science, University of Reading, Reading, RG6 6ET, UK
[*]Now at Met Office, Exeter, UK

**Correspondence:** Suzanne Gray (s.l.gray@reading.ac.uk)

**Abstract.** Human activity in the Arctic is increasing as new regions become accessible, with a consequent need for improved understanding of hazardous weather there. Arctic cyclones are the major weather systems affecting the Arctic environment during summer, including the sea ice distribution. Meso- to synoptic-scale tropopause polar vortices (TPVs) frequently occur in polar regions and are a proposed mechanism for Arctic cyclone genesis and intensification. However, while the importance
of pre-existing tropopause-level features for cyclone development, and as an integral part of the three-dimensional mature cyclone structure, is well established in the mid-latitudes, evidence of the importance of pre-existing TPVs for Arctic cyclone development is mainly limited to a few case studies. Here we examine the extent to which Arctic cyclone growth is coupled to TPVs by analysing a climatology of summer Arctic cyclones and TPVs produced by tracking both features in the latest ECMWF reanalysis (ERA5).

The annual counts of Arctic cyclones and TPVs are significantly correlated for features with genesis either within or outside the Arctic, implying that TPVs have a role in the development of Arctic cyclones. However, only about one third of Arctic cyclones have their genesis or intensify while a TPV of Arctic origin is (instantaneously) within about twice the Rossby radius of the cyclone centre. Consistent with the different track densities of the full sets of Arctic cyclones and TPVs, cyclones with TPVs within range throughout their intensification phase (matched cyclones) track preferentially over the Arctic Ocean along
the North American coastline and Canadian Archipelago. In contrast, cyclones intensifying distant from any TPV (unmatched cyclones) track preferentially along the north coast of Eurasia. Composite analysis reveals the presence of a distinct relative vorticity maximum at, and above, the tropopause level associated with the TPV throughout the intensification period for matched cyclones and that these cyclones have a reduced upstream tilt compared to unmatched cyclones. Interaction of cyclones with TPVs has implications for the predictability of Arctic weather, given the long lifetime, but relatively small spatial scale of
TPVs compared with the density of the polar observation network.

## 1  Introduction

Arctic cyclones are typically defined as synoptic-scale cyclones developing within, or moving into, the Arctic region. In summer, Arctic cyclones are larger than the polar meso-cyclones that are common during Arctic winter, of which polar lows are the most intense subset. Climate models project continued summer sea ice reductions in the Arctic with nearly ice-free conditions

by the middle of this century (Overland and Wang, 2013). Human activity in the Arctic, such as from shipping and offshore operations, is expected to increase as new regions become accessible (Stephenson et al., 2013) with a consequent need for reliable weather forecasts, particularly of hazardous weather. Arctic cyclones also locally affect sea ice cover through surface fluxes and wind forcing, with cyclones appearing to slow the general day-to-day decline in concentration during the summer months (Finocchio et al., 2020; Schreiber and Serreze, 2020; Lukovich et al., 2021). Summertime Arctic cyclone activity is positively linked to the strength of the land-sea thermal contrast along the Arctic coastline termed the Arctic Frontal Zone (e.g. Crawford and Serreze, 2016) and projected increases in dynamical intensity and frequency of Arctic cyclones in summer are associated with enhancement of this zone (Day and Hodges, 2018). In contrast, Arctic cyclone intensity is projected to decrease in the winter (Day et al., 2018). Wickström et al. (2020) link recent historical (1979-2016) significant trends in winter cyclone densities in the Svalbard and Barents Sea regions to a change to a more meridional North Atlantic storm track. An ongoing World Meteorological Organisation World Weather Research Programme (the Polar Prediction Project) was developed in recognition of the importance of improved weather and environmental prediction services for the Polar Regions. The meso- to synoptic-scale tropopause-based coherent vortices (called tropopause polar vortices, TPVs) frequently observed in polar regions (Hakim and Canavan, 2005; Cavallo and Hakim, 2009) are hypothesised to play a central role in Arctic cyclone genesis and intensification (Tao et al., 2017a; Yamagami et al., 2017, 2018a). In this paper we present a climatology of summer Arctic cyclones and TPVs by tracking features in the latest global reanalysis of the European Centre for Medium-Range Weather Forecasts (ECMWF), ERA5, and deduce the role of TPVs in the initiation, intensification and structure of Arctic cyclones.

Arctic cyclones are often distinguished from their extratropical cousins by a simple latitude threshold. The climatological characteristics of Arctic cyclones (and indeed all cyclones) are somewhat dependent on the identification and tracking tool used for analysis as well as the dataset to which it is applied (typically a reanalysis, see Screen et al. (2018) for a review). Using the National Centers for Environmental Prediction-National Center for Atmospheric Research (NCEP-NCAR) reanalysis (from 1948–2002) and tracking mean sea level pressure (MSLP) anomalies, Zhang et al. (2004) found the number of Arctic cyclones (cyclones north of $60°$) increased slightly from winter to summer (from about 65 to 75 per month). The summer cyclones had slightly longer durations (exceeding 40 hours on average) but reduced intensity, measured by MSLP, compared to the winter cyclones. Cyclones generated locally in the Arctic region were weaker than those tracking into it from the mid-latitudes. Enhanced model resolution (and improved data assimilation) in reanalyses leads to an improved representation of cyclones north of $55°$N with deeper central pressures, faster deepening, stronger winds and up to a 40 % increase in summer (June–October) cyclone numbers when comparing the Arctic System reanalysis, which has 30 km horizontal grid spacing, with coarser resolution reanalyses (Tilinina et al., 2014). However, as expected, this difference in numbers is most marked for weaker cyclones (central MSLPs exceeding 980 hPa). Also when comparing the reanalyses for just the Arctic Ocean region, the differences between them were much reduced with the Arctic System reanalysis having only 9% more cyclones than the average of the other reanalyses considered. The analysed characteristics of Arctic cyclones depend on the feature identification method and post-tracking filtering. Cyclone frequency was found to be higher in winter than summer when identifying synoptic-scale cyclones using 850 hPa relative vorticity ($\xi_{850}$), but similar when identifying them using MSLP using the same tracking algorithm (Vessey et al., 2020). These differences can exceed those arising from the use of different

reanalyses. When identifying using $\xi_{850}$, about 100 synoptic-scale Arctic cyclones (cyclones north of 65°N) occurred each summer season (June–August) and 120 each winter season (December–February) with about half of these cyclones having their genesis in the Arctic region and the other half tracking into it from the mid-latitudes in both seasons. In contrast, only about 65 cyclones occurred each summer and winter season when identifying them using MSLP. In consrast to the range in cyclone numbers, previous studies generally agree on the geographical characteristics of Arctic cyclones. In summer cyclones tend to track along the Arctic Frontal Zone region, particularly along the northern Eurasian coastline, and into the Arctic Ocean (e.g., Vessey et al., 2020). The east coast of Greenland and the northern Canadian coastline are also regions of cyclogenesis.

Tropopause polar vortices (TPVs) are defined by Cavallo and Hakim (2009) as long-lived coherent vortices associated with a meso- to synoptic-scale depression of the tropopause (most often less than 1000 km in radius). These can last longer than a month (Hakim and Canavan, 2005) and can be isolated from other features at tropopause level. Their importance is associated with their ability to spin up low-level disturbances through baroclinic interaction, with their longevity offering up potential for enhanced predictability. Note that TPVs are distinct from the much larger-scale *tropospheric polar vortex* and stratospheric winter polar vortex that are both associated with the reservoir of higher potential vorticity (PV) values occupying high latitudes (see Waugh et al. (2017) for a description of these planetary-scale vortices). The PV gradient is often sharp on the edge of the tropospheric polar vortex, where isentropic surfaces intersect the tropopause, and the westerly jet streams are located on these sharp gradients. One source of confusion with the terminology used in the literature arises because the tropospheric polar vortex weakens (and its area shrinks) between winter and summer to such an extent that it eventually breaks up into a number of smaller cut-off PV features at tropopause level. After this happens, the individual features have been referred to as both polar vortices and TPVs, although they may be quite large in scale (especially in early summer). Eventually, the larger-scale tropospheric polar vortex is re-established in autumn as a result of increasing net radiative cooling re-building the PV reservoir.

Cyclonic TPVs are associated with a lowered tropopause and therefore larger values of PV where stratospheric air is at lower altitude than usual (see e.g., Cavallo and Hakim, 2010). The PV feature may be cut-off from other PV features on isentropic surfaces intersecting the tropopause, or it may be on the edge of a large-scale Rossby wave trough. Either way, the dynamic tropopause surface has a local minimum in altitude describing the centre of the TPV. Many studies, including ours, use the surface where the PV equals 2 PVU (potential vorticity units, $1\,\mathrm{PVU} = 10^{-6}\mathrm{m}^2\mathrm{s}^{-1}\mathrm{K}\,\mathrm{kg}^{-1}$) to define the dynamic tropopause; negative anomalies of potential temperature, $\theta$, on the 2 PVU surface are used to identify TPVs. Note that $\theta$ increases with height throughout most of the atmosphere, indicating stable stratification, so a TPV associated with a minimum in tropopause height must naturally be associated with a minimum in $\theta$ relative to its surroundings on the tropopause. In the Northern Hemisphere, the positive PV anomaly characterising the TPV is associated with positive $\xi$, and anticlockwise relative motion, as well as a local maximum in static stability ($\partial\theta/\partial z$). If the TPV is an isolated PV feature, then this is achieved by vertically bunching isentropic surfaces towards the centre of the PV anomaly (just above the tropopause), resulting in regions with lower static stability above and below (Hoskins et al., 1985). The isentropic surfaces dip down in the lower stratosphere above the PV anomaly and therefore there is a positive $\theta$-anomaly ($\theta'$) there (and also high temperature anomaly relative to surroundings on each pressure surface). In the upper troposphere below the anomaly, the isentropic surfaces bow upwards and there is a negative $\theta'$ that is often described as a "cold core".

In Cavallo and Hakim (2010) composite analysis of TPVs forming in the Canadian Arctic region over a two-year period, simulated using the Weather Research and Forecasting (WRF) model, revealed peak temperature anomalies of -8.5 K and +5.5 K in the troposphere and above the tropopause (at around 250 hPa), respectively, with negative $\theta$-anomalies on the tropopause. The positive PV anomaly associated with the composite TPV approached 4 PVU above the lowered tropopause, peaking at the level of the composite background tropopause. Analysis of the tendency in anomalous PV due to diabatic

processes demonstrated that this positive PV anomaly was enhanced primarily by radiative processes (anomalously strong cooling at the tropopause level and anomalously weak cooling above) with a weaker compensation, at the tropopause level, from latent heating. Cavallo and Hakim (2013) further showed, using idealised simulations, that TPV intensification is mainly due to anomalous longwave radiative cooling associated with the tropopause depression and displacement of the very low stratospheric specific humidity to much lower altitudes. The main climatological genesis region of TPVs is the Canadian

archipelago and northern Baffin bay region, although genesis also occurs along and north of the Eurasian and North American coastline with a secondary maxima over the Kara sea (see Fig. 1 of Cavallo and Hakim (2009)). Despite the documented existence of some very long-lived cyclonic TPVs (e.g., Hakim and Canavan, 2005; Lillo et al., 2021) suggesting long tracks, the climatological lysis regions in Cavallo and Hakim (2009) lie immediately downstream of the genesis regions.

  While forecast skill in the Arctic is comparable to that in the Northern hemisphere mid-latitudes, despite increased analysis

uncertainty (Jung and Matsueda, 2016), forecast skill can be poorer for Arctic cyclones than for mid-latitude cyclones (Capute and Torn, 2021). From analysis of ten "extraordinary" (large and intense) Arctic cyclones represented by ensemble forecasts from five operational centres, Yamagami et al. (2018b) found average location errors of about 470 km and central MSLP errors of 6–11 hPa for lead times of 2.5–4.5 days before maturity. The "Great Arctic cyclone of 2012" was notable for its exceptionally long lifetime (12 days) as well as intensity (minimum central MSLP of 966 hPa) (Simmonds and Rudeva, 2012). While the

rapid intensification of this cyclone arose from lower-tropospheric baroclinicity (inferred from the Eady growth rate), a co-located TPV was important for the development of the surface cyclone. Analysis of the predictability of this cyclone revealed that this was increased by accurate prediction of upper-level, particularly temperature, features (Yamagami et al., 2018a). This cyclone formed due to the merging of both the upper-level warm cores (diagnosed as 250 hPa temperature anomalies) and surface cyclones of a mid-latitude and Arctic cyclone, and the timing and southwards movement of a "polar vortex" (diagnosed

using 300 hPa geopotential height minimum) was crucial to the position and development of the merged cyclone; Yamagami et al. (2018a) noted that their definition of a polar vortex was similar to the TPVs defined by Cavallo and Hakim (2010). The geopotential height minimum on a pressure surface is associated with positive vorticity through geostrophic balance. The "warm cores" defined on pressure surfaces are positive $\theta$-anomalies and, as explained above, arise above TPVs because $\theta$-surfaces must dip downwards above the positive PV anomaly. Consistent with the findings of Yamagami et al. (2018a),

Yamazaki et al. (2015) found that assimilation of additional radiosonde observations was crucial for accurate forecasts of this cyclone. Tao et al. (2017a) also found that a polar vortex was important for the intensification of this cyclone, specifically through its role in intensifying the upper-tropospheric jet. A similarly exceptional Arctic cyclone developed in 2016, lasting for more than one month. This cyclone was maintained through multiple merging of cyclones and their associated "warm-cored polar vortices" (i.e., TPVs) (Yamagami et al., 2017; Ishiyama and Tanaka, 2021).

It is not surprising that TPVs can have an important role in the formation and intensification of Arctic cyclones. Pre-existing upper-level PV anomalies, such as associated with upper-level troughs or smaller-scale jet steaks, have long been recognised as being able to initiate baroclinic growth in the mid-latitudes if they move over a low-level baroclinic zone (as classically described in Sect. 6e of Hoskins et al. (1985)). Petterssen and Smebye (1971) introduced the term "type B cyclogenesis" to describe the baroclinic mechanism of growth dominated by a finite amplitude upper-level precursor disturbance. This con-

trasts with "type A cyclogenesis" where a low-level wave develops on a baroclinic zone, initially without much upper-level disturbance (although one may later develop). Deveson et al. (2002) later extended this classification scheme to include type C cyclogenesis, characterised by strong mid-level latent heat release. Considering nearly 700 cyclones in the North Atlantic region, Gray and Dacre (2006) diagnosed roughly equal numbers type A, B and C cyclones (30, 38 and 32 % respectively) with type B cyclogenesis dominating for the Gulf Stream region off the East Coast USA. While the importance of upper-

level PV anomalies for cyclone development, as well as their almost ubiquitous existence as part of the three-dimensional mature cyclone structure (e.g. Čampa and Wernli, 2012), is well established in the mid-latitudes, evidence of the importance of pre-existing TPVs for Arctic cyclone development is more limited. Simmonds and Rudeva (2014) found that, at their time of maximum intensity, all but 6 of a set of 60 Arctic cyclones (the 5 most intense in each calendar month over a 30 year period) had a significant cyclonic feature in 300-hPa geopotential height within 555 km; however, these upper-level cyclonic

features were not explicitly linked to pre-existing TPVs. Tanaka et al. (2012) diagnosed an untilted structure with a vortex tube extending from the surface to the lower stratosphere at the mature stage of three Arctic cyclone case studies. Pre-existing TPVs have been associated with a few exceptionally large, intense and long-lived Arctic cyclones in case studies (Yamagami et al., 2018a, 2017), but their association more generally with typical cyclones is not known. In this study we explore this research gap by determining how often pre-existing TPVs are important for Arctic cyclone development over 40 extended summers

(May–September). The following research questions are addressed:

   – What are the statistical characteristics of Arctic cyclone tracks and how do they compare to those of TPVs?

   – What is the role of TPVs in the initiation and intensification of Arctic cyclones?

   – How is the structure of Arctic cyclones modified by interaction with TPVs?

   The paper continues as follows. The methods are described in Sect. 2, beginning with the identification and feature tracking

of Arctic cyclones and TPVs in 40 extended summers of ECMWF fifth generation (ERA5) reanalysis data (Hersbach et al., 2020) and then defining the method for matching Arctic cyclone and TPV features. Section 3 begins with quantification of the climatological characteristics of Arctic cyclone and TPV tracks before moving on to the spatial co-location of these features at the times of Arctic cyclone genesis, maximum growth rate and maximum intensity. Spatial composites are contrasted for two sets of Arctic cyclones, namely matched or unmatched with TPVs, at the time of maximum intensity, the earlier time of

maximum growth rate and for two days before that. The composite structures are used to deduce the influences of TPVs on Arctic cyclone development and compared with those of mid-latitude cyclones. Section 4 contains the conclusions.

## 2 Methodology

### 2.1 Identification and feature tracking

Arctic cyclones and TPVs are identified in the ERA5 dataset, the highest resolution reanalysis dataset available from ECMWF. ERA5 was produced using cycle Cy41r2 of ECMWF's integrated forecast system which was operational from 8 March to 21 November 2016. The IFS model was integrated with a horizontal spectral truncation of TL639 with 137 terrain-following hybrid-pressure levels up to 80 km. Three-hourly data on a N320 Gaussian grid (approximate meridional grid spacing of $0.281°$ or 31 km) was used for the extended summer season, May–September inclusive, from 1979–2018.

Arctic cyclones and TPVs were identified using maxima of $\xi_{850}$, and minima of $\theta$ on the dynamic tropopause (defined as the surface with PV of 2 PVU), $\theta_{2PVU}$, respectively. The TRACK algorithm (version 1.5.2) was used to track the identified features (Hodges, 1995, 1999). After tracking, associated MSLP minima (if they existed) were determined for each Arctic cyclone track point. The associated MSLP minimum was diagnosed as that closest, within a $5°$ radius, using B-spline interpolation and a steepest descent minimization, and using the $\xi$ track location as the starting point. For tracking, mid-latitude cyclones are typically identified using either $\xi_{850}$ or MSLP (e.g. Neu et al. (2013)). The 850 hPa level is used for $\xi$ to avoid strong influences from boundary layer processes and orography. As described in Sect. 1, both fields were used in a recent comparison of Arctic cyclones tracked using different global reanalyses that used the same tracking algorithm as used in this study (Vessey et al., 2020). The authors found that more Arctic cyclones were identified using $\xi_{850}$ than using MSLP due to the identification of smaller-scale systems using $\xi_{850}$. In the summer about 50 % more cyclones were identified using $\xi_{850}$, whereas in the winter the values nearly doubled. Note though that these findings are likely to be somewhat dependent on the spatial filtering before tracking and the post-tracking filtering. Far fewer studies have tracked TPVs and our identification of these systems using $\theta_{2PVU}$ minima follows that of Cavallo and Hakim (2010).

Spatial filtering of the fields is used prior to feature identification and tracking to remove both the planetary-scale background flow and smaller mesoscale features and to focus on the scales of interest; note this filtering is different to the amplitude-based filtering used by Cavallo and Hakim (2010). Spectral filtering was applied to retain features with wavenumbers in the range T5–T63 (where T$N$ refers to triangular truncation of the spherical harmonics) for both fields to yield anomalies. Lander and Hoskins (1997) argue that $\pi a/N$ (where $a$ is the radius of the Earth) is a good estimate of the smallest resolved scale for circular features (it is smaller for wave-like features), which is approximately 320 km for $N = 63$. This does not mean that there is a sharp cut-off in the scale of the features represented (such that smaller features are absent); they will tend to be smoothed, removing the small-scale noise that can result in multiple centres. Centres can be resolved (i.e., unambiguously distinguished) if their separation is greater than 320 km. In terms of equivalence to a grid-point model, the usual guidance is that a minimum of 5–6 grid points are needed to partially resolve a feature – so this spectral resolution is equivalent to a grid-point model with spacing of about 60 km. Cavallo and Hakim (2010) find that the vast majority of TPVs (after filtering that was designed to isolate well-resolved vortices) have radii exceeding 200 km, and so diameters exceeding 400 km. Hence, these features would be represented with the T63 upper limit of filtering used in our study, justifying the truncation used. Data were then projected onto a $200{\times}200$ grid on a polar stereographic projection for feature identification before the feature points were

mapped back to the sphere for tracking. Note that because the atmospheric evolution is close to adiabatic and frictionless, to a good approximation $\theta$ is advected conservatively on the 2 PVU surface and therefore no maxima or minima can be created in $\theta$ that are not in the initial conditions. Hence we track minima in $\theta$-*anomalies*, defined by filtering out the planetary scale $\theta$-field with $N < 5$, which can be generated by equatorward displacement of air with low $\theta$ values (Anderson et al. (2003) discuss the sensitivity of tracking to the form of background field removal). In contrast, by definition $\xi$ is already an anomaly relative to the planetary vorticity distribution and so the weak background field does not need to be removed; however, this removal is done for consistency. Preliminary case study analysis in which different spectral filtering ranges were compared (not shown) showed the chosen filtering retained features of interest while removing smaller mesoscale features, such as fronts, and smaller TPV features. For tracking several thresholds were employed. For identification for the tracking, Arctic cyclones were identified in $\xi_{850}$ using a minimum value of $10^{-5}\text{s}^{-1}$ for the local maxima and TPVs were identified in $\theta_{2\text{PVU}}$ using a maximum value of -1 K for the local minima. Following the tracking, tracks were retained if they existed for more than 1 day (8 time steps) and the track distance exceeded 1000 km (to focus on mobile systems). Spatial statistics were computed from the tracks using spherical kernels (Hodges, 1996).

## 2.2  Post-processing of tracked features

Tracks for both TPV and Arctic cyclones were partitioned into those that formed within the Arctic region (defined here as North of 65°N, following Vessey et al. (2020)) and those that formed further south and moved into the Arctic. To determine the role of TPVs in the initiation and intensification of Arctic cyclones, Arctic cyclones were first matched to TPVs independently at the times of genesis, maximum growth rate and maximum intensity of the Arctic cyclones. The matching was performed for horizontal separation geodesic distances (matching radii) of 1° to 10°. The genesis time of features is defined as the first time at which they are identified (above threshold intensity). The time of maximum growth rate of cyclones is defined at the start of the three-hour period over which the filtered $\xi_{850}$ has the maximum increase along the track. The time of cyclone maximum intensity is defined by the filtered $\xi_{850}$ maximum along the track. Next, Arctic cyclones were matched to TPVs with which they had a sustained association as follows.

Two subsets of Arctic cyclones are defined: those that are matched and those that are unmatched to a TPV during their intensification period (termed matched and unmatched cyclones). The matching procedure is demonstrated in Fig. 1 for a sample cyclone, selected because it is one of the cyclones that is related to the extreme Arctic cyclone of 2016; it is the same cyclone as that shown in Fig. 1(f) of Yamagami et al. (2017) originating over Scandinavia, and the intensities of the matched TPV and cyclone along the track are shown in Fig. 1(a) together with the MSLP field at the time of maximum intensity of the cyclone. Matched cyclones are defined as cyclones that are within 5° of an Arctic origin TPV at the time of their maximum intensity and within 10° of the same TPV at the time of their maximum growth rate (thus if a cyclone interacts with a different TPV at its time of maximum intensity and growth rate it will not be considered as matched to a TPV). These radii are shown in Fig. 1(b) for the sample cyclone by transparent red and blue circles, and the TPV locations at the corresponding times can be seen to be within the matching radii from the track symbols which are coloured according to the timing along the two tracks. We note that in the sample case the period between the times of maximum growth rate and maximum intensity is rather short

(nine hours), though this is not necessarily typical. The separation between the cyclone and TPV at the time of maximum intensity is only just within the radial limit and we speculate that this is because the interacting features are rather large here – the cyclone has a small low centre embedded in a much larger low pressure region. The interaction between the cyclone and TPV is associated with a very clear change in direction of the tracked TPV motion. This sample case presents the low-level cyclogenesis over the Russian Arctic Frontal Zone and later interaction with TPV while right over the pole. The cyclone did

not last long after this interaction, while the TPV continued to orbit for many days over the Canadian side of the Arctic.

The matching is performed iteratively: matching is first applied at the time of maximum intensity yielding possible matched cyclones and TPVs, and then the constraint on matching at the time of maximum growth rate is applied to those systems; more than one cyclone can be matched to the same TPV as this provides the best sample of cyclones. Unmatched cyclones are defined as cyclones that are further than $10°$ from a TPV at both their time of their maximum growth rate and time of

240 their maximum intensity. These feature separation thresholds were chosen assuming that the Arctic cyclones evolved from a tilted vertical structure to a less tilted vertically-aligned structure at maturity (maximum intensity) as expected for baroclinic growth and shown in case studies for cyclones interacting with TPVs (e.g., see Fig. 4 of Tao et al., 2017a). Additionally, both matched and unmatched cyclones must also exist for at least two days prior to their maximum growth rate (the location of the sample cyclone at this time is indicated by the black transparent square in Fig. 1(b)) and achieve their maximum intensity

in the Arctic. This constraint enables consistent investigation of the structure of Arctic cyclones at one and two days prior to maximum growth rate. Note that the TPV could be located in any direction relative to the Arctic cyclone centre and simply matches if it is within a threshold radius. Unmatched cyclones must have no TPV within a specified radius in any direction from the cyclone centre.

The aim in defining the criteria for matched and unmatched cyclones is to distinguish between cyclones that are very likely

and very unlikely, respectively, to interact with a TPV. The Rossby deformation radius in the Arctic is estimated to be at most 500 km (for a tropopause at 7 km and Coriolis parameter of about $1.5 \times 10^{-4}$ s$^{-1}$). The length scale characteristic of the velocity induced by PV anomalies depends on the shape of the PV structures interacting and the nature of interaction, varying between the Rossby deformation radius for point vortices and $1/k$ for large-scale sinusoidal PV waves (where $k$ is wavenumber). Hence, by using a matching criterion requiring separation between a cyclone and TPV of less than about twice

the Rossby deformation radius ($10°$) at the time of maximum growth rate we are allowing interaction for a range of shape of the disturbances from point vortices to waves. Similarly, by requiring that unmatched cyclones are further apart than twice the Rossby deformation radius, even at their maximum intensity time, when systems are more vertically stacked, our unmatched cyclones are very unlikely to be interacting strongly with a TPV. We have also considered more restrictive criteria for matching ($5°$ at maximum growth rate time and $2°$ at maximum intensity time) and this does not affect the overall conclusions of the

work.

Composite matched and unmatched cyclones were generated using the 200 most intense cyclones (defined by the filtered $\xi_{850}$ at the time of maximum intensity) from the matched and unmatched sets, respectively. Composites were produced using superimposed epoch analysis for the time of maximum growth rate of the cyclones, one and two days prior to this time, and the time of maximum intensity of the cyclones (note that lifecycle analysis (not shown) revealed negligible difference between the

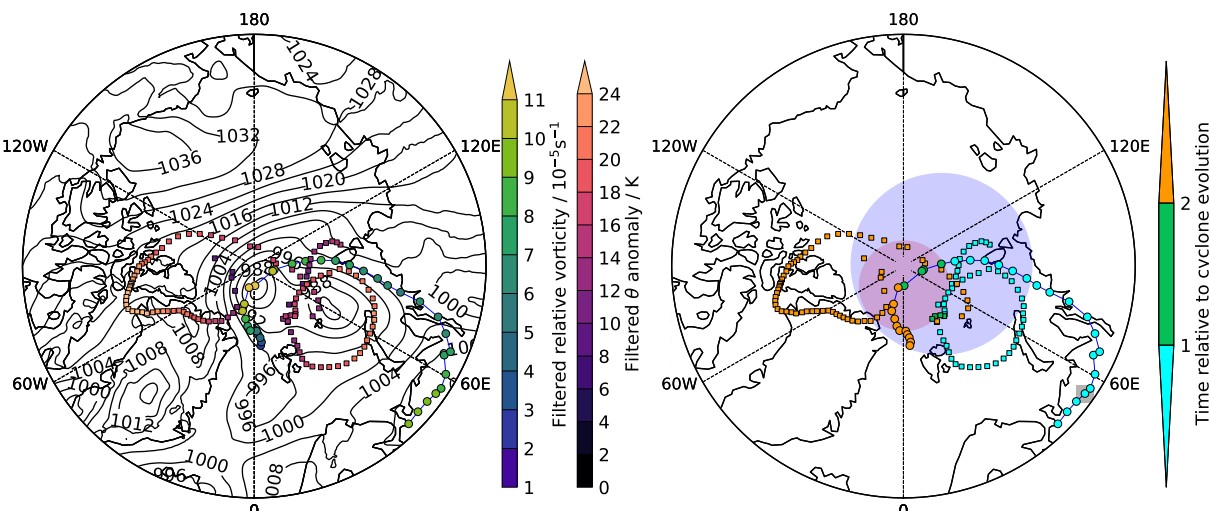

**Figure 1.** Example of matched cyclone and TPV with circle markers linked by lines and (unlinked) square markers respectively showing the tracks of the cyclone and TPV at three-hourly intervals: (a) intensity of tracks with MSLP at the time of the maximum intensity of the cyclone (15 UTC 30 August, defined by filtered $\xi_{850}$) in black contours (4 hPa interval) and (b) timing of tracks (green fill indicates times from that of maximum growth rate (labelled as '1' on the colourbar) to that of maximum intensity (labelled '2') inclusive and blue and orange fill indicates times prior to that of maximum growth rate and post that of maximum intensity, respectively) with blue transparent circle indicating 10° radius around the cyclone at its time of maximum growth rate, red transparent circle indicating 5° radius around the cyclone at its time of maximum intensity, and black transparent square indicating the location of the cyclone two days prior to its time of maximum growth rate.

timing of the MSLP minimum and relative vorticity (and wind speed) maxima). The composites were generated by first setting up a rectangular grid centred on the equator. This grid was then rotated to the centre of each matched or unmatched Arctic cyclone at the time required (e.g., its time of maximum intensity) and orientated relative to the direction of the movement of that cyclone so that all cyclones are orientated in the same direction relative to their motion. Data were then sampled to the grid and finally averaged over the cyclones to produce the composites. The number of cyclones for compositing was chosen as

a compromise between choosing too few cyclones, such that the composite fields plotted may not be representative, and too many, such that cyclone features may be smeared out by large differences between the most and least intense systems.

     The tilt structure of the composite cyclones, defined by the vertical structure of $\xi$ maxima on selected pressure levels, was calculated based on the method used in Bengtsson et al. (2009). The tilt for each of the 200 matched (or unmatched) cyclones was calculated recursively by identifying the maxima at each level using the maxima at the previous level as a starting point

for the search. The search was performed starting at 900 hPa (the bottom of the profile) in the filtered data and continuing until 50 hPa, using a search radius of 3° (though similar results were obtained using a search radius of 5° (not shown)). A steepest ascent and B-spline interpolation method was used to determine the maxima. These maxima were then projected onto the cyclone motion direction so that the tilt was relative to this direction (in spherical geometry) and the position relative to the 850 hPa centre (the level at which the cyclone motion direction is defined) determined as a geodesic angle. The tilts were

then adjusted to be relative to the 900 hPa location. Finally, the radii were averaged at each level over the set of matched (or unmatched) cyclones relative to the time of maximum growth rate to yield the three-dimensional composite structure. Note that occasionally a sufficiently close $\xi$ maximum could not be found at a level; this becomes more likely at higher levels. The composite tilt structure was calculated from two days before to two days after the time of maximum growth rate. While each cyclone must exist from at least two days before the time of its maximum growth rate due to the criteria for selection of the matched and unmatched cyclones described above, there is no requirement for their existence after this time. Hence, a small number of cyclones do not have identified tracks for the full two days after their time of maximum growth rate.

## 3   Results

### 3.1   Climatological characteristics of TPVs and Arctic cyclones

The genesis density of tracked Arctic cyclones and TPVs is shown in Fig. 2 (top and bottom row, respectively) both for all systems and split into systems with genesis within and outside the Arctic. Considering first the Arctic cyclones, there are clear enhanced regions of genesis along several coastlines around the Arctic. The strongest genesis region lies along the east coast of Greenland. Arctic cyclones with genesis outside of the Arctic have preferred genesis regions over the Canadian Rockies and southern Scandinavia (Fig. 2(c)). These preferred genesis regions are consistent with previous studies performed using different reanalyses (e.g., Vessey et al., 2020). The genesis locations of TPVs are less localised than those of Arctic cyclones (Fig. 2(d)). However, there are still a few notable areas of enhanced genesis density. The strongest of these areas is located over northern Canada and the Canadian Arctic Archipelago (as also found by Cavallo and Hakim (2009)). Other density maxima are located over the North Pole and northern Siberia. It is rare for TPVs to have genesis regions outside of the Arctic (Fig. 2(f)), although they can migrate out of the Arctic consistent with the negative $\theta$-anomalies (on the tropopause) tending to be associated with equatorward displacement.

The track density of tracked Arctic cyclones and TPVs is shown in Fig. 3 organised in the same way as for genesis density. The track density plots are smoother than those of genesis density and the track density of Arctic cyclones is much more evenly spread across the Arctic Ocean than the genesis density. So, although the Arctic cyclones tend to be preferentially first identified near the coastlines, many subsequently track towards the North Pole. Nevertheless, there is enhanced track density over northern Russia, which is the main area influenced by Arctic cyclones with genesis regions both within and outside the Arctic. The only area of the Arctic where Arctic cyclones are not observed is across the centre of Greenland; this gap is a consequence of the high altitude of the Greenland plateau which means that it lies above the 850 hPa surface used to identify cyclones leading to disruption of the tracking. Unlike for Arctic cyclones, the track density plots for the TPVs (Fig. 3d–f) are somewhat similar to those for genesis density. This result is consistent with the finding of Cavallo and Hakim (2009) that the TPV lysis regions are immediately downstream of the genesis regions. Figure 3(d) (for all TPVs) shows a relatively strong density maximum over the Canadian Arctic Archipelago, though slightly further east than for genesis density; the rest of the tracks are found along the Arctic coastlines, at an approximate latitude of 70°N. This track density map is consistent in structure with those for intensifying TPV occurrence shown in Figs. 1 of Cavallo and Hakim (2009) and Cavallo and Hakim (2010).

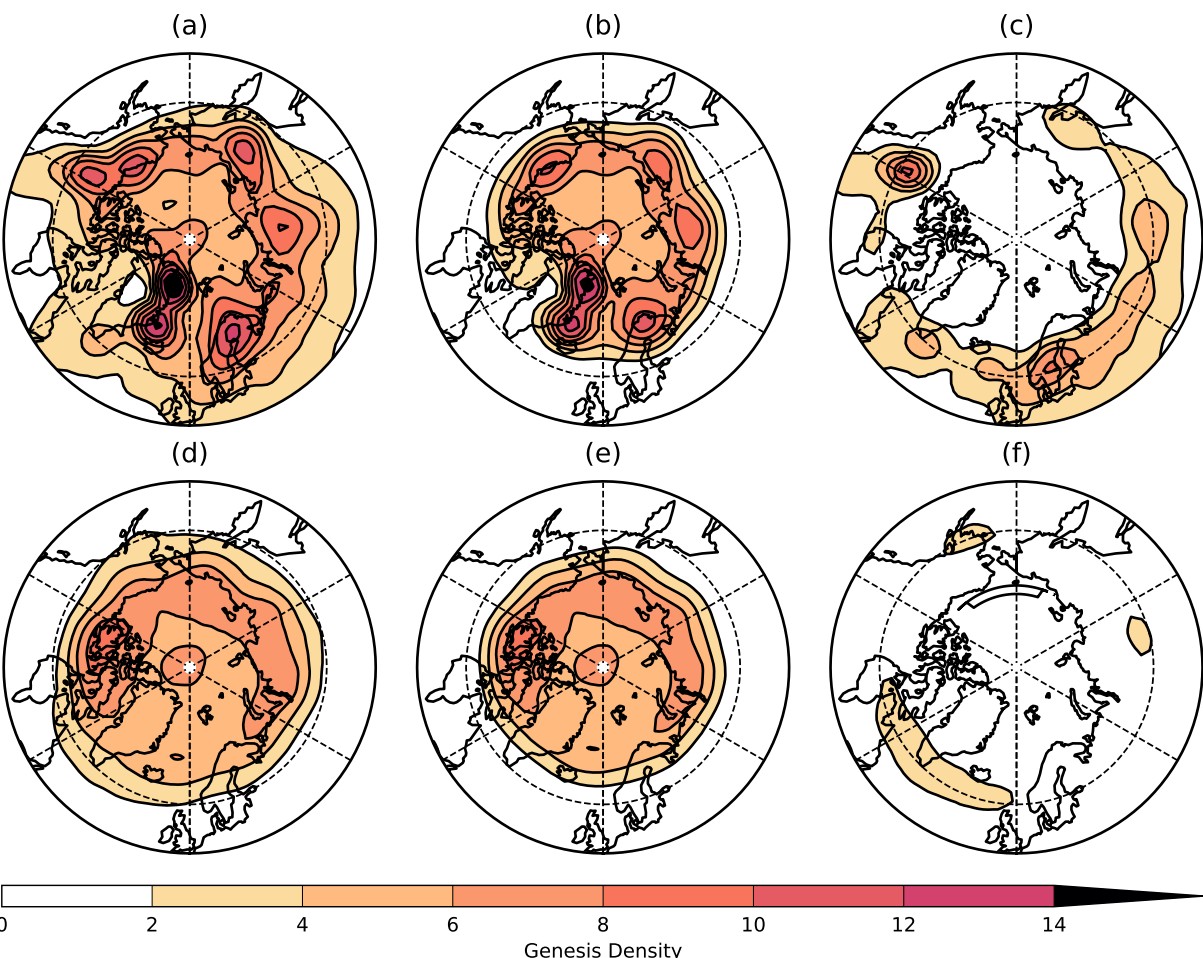

**Figure 2.** Genesis density of Arctic cyclones for (a) all cases, (b) Arctic genesis only and (c) non-Arctic genesis. Genesis density of TPVs for (d) all cases, (e) Arctic genesis only and (f) non-Arctic genesis. Data spans the extended summer season between 1979 and 2018. Units are number per unit area per season where the unit area is equivalent to a five degree spherical cap ($\sim 10^6$ km$^2$). The total number of tracks is (a) 12155, (b) 6822, (c) 5333, (d) 8339, (e) 6288, and (f) 2051. Maps are orientated with 0° longitude at the bottom.

Notably, there is a relatively lower track density in the central Arctic Ocean. In particular, very few TPVs with genesis outside the Arctic make it into the central Arctic region (Fig. 3(f)). The genesis density and track density maps were also considered

for all Arctic cyclones and Arctic genesis TPVs for each month individually during the extended-summer season (not shown). As expected, the maps are less smooth when considering the months individually, compared with the extended-summer period, and there is also some month-to-month variability. However, the basic features of the maps, as described above, are also present for each month.

The distributions of lifetimes and intensities of the tracked Arctic cyclones and TPVs are shown in Fig. 4 (left and right

panels, respectively); note that in all panels the means of the two distributions plotted are significantly different at the 95 % level

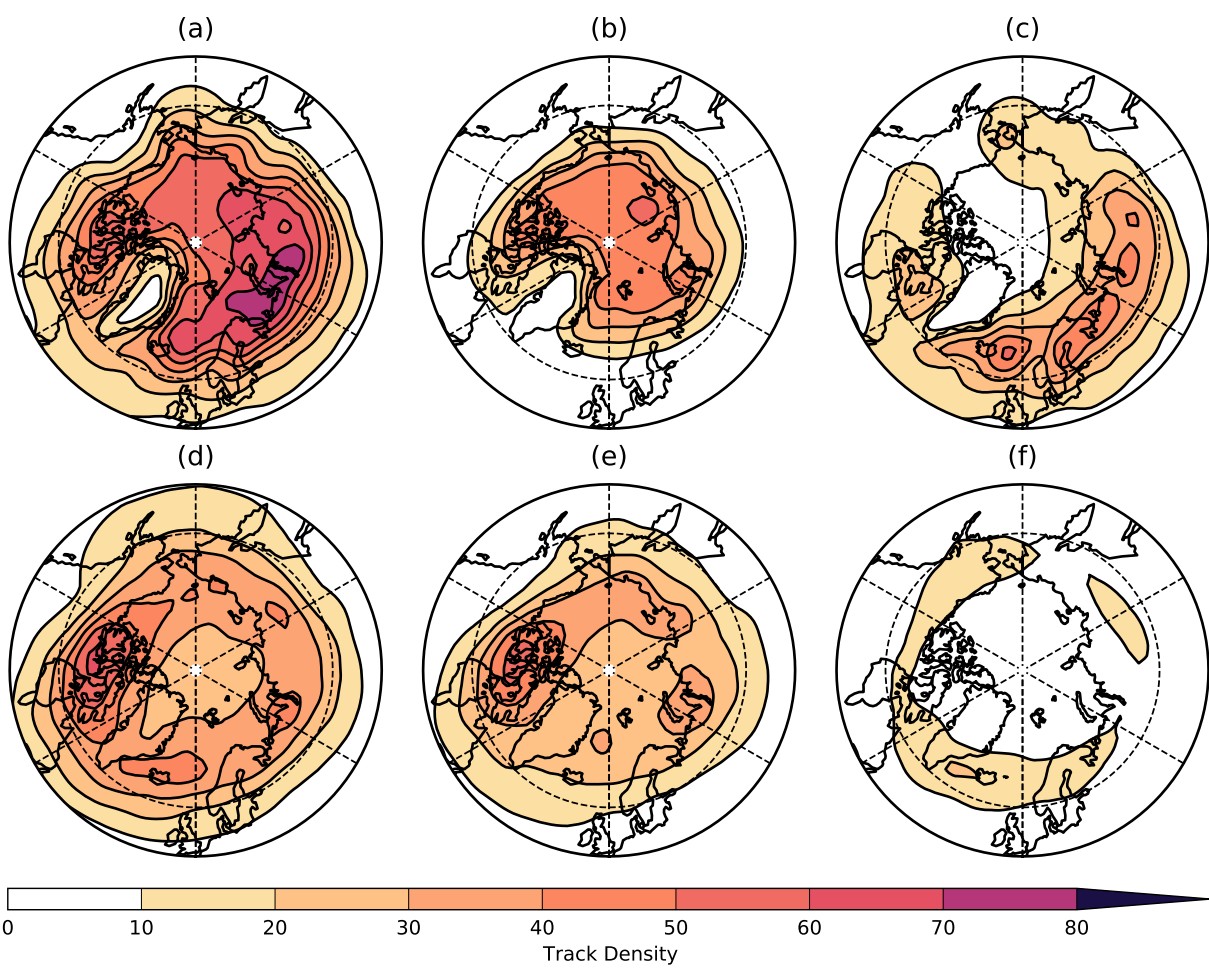

**Figure 3.** As for Fig. 2 but for track density. (a, b, c) for Arctic cyclones and (d, e, f) for TPVs.

according to a two-sided Welch's t-test (without assuming equal variance). Lifetime distributions are compared for the Arctic cyclones and TPVs (Fig. 4(a)) and the Arctic and non-Arctic genesis cyclones (Fig. 4(c)) and TPVs (Fig. 4(e)), respectively. The modal lifetime of Arctic cyclones of 2–3 days exceeds that of TPVs (1–2 days), but the number of Arctic cyclones drops off more steeply with lifetime than for TPVs, and the Arctic cyclone lifetime distribution has a shorter and sparser tail than that of
325 TPVs (Fig. 4(a)). Consequently the mean TPV lifetime is slightly longer than that of Arctic cyclones (5.0 compared to 4.4 days). Some exceptionally long-lived TPVs exist with the maximum lifetime found being 43 days. It is possible that exceptionally long lifetimes could result from the tracking algorithm erroneously connecting tracks associated with two different TPVs. However, more detailed examination of some of these tracks suggests that they are reliable; for example, Hakim and Canavan (2005) also found that TPV lifetimes could exceed one month. We speculate that some of these long-lived TPVs are the remnants of
330 the tropospheric polar vortex which becomes much smaller moving into summer and typically breaks into a few smaller PV features at some point. These TPVs are long lived because they are coherent stronger vortices which resist the relatively weak

large-scale strain and also because longwave cooling maintains the positive PV anomaly where the tropopause is lower. Arctic cyclones are more likely to be long-lived if they track into the Arctic region than if they have their genesis there (Fig. 4(c), mean lifetimes 4.9 and 3.9 days, respectively). Conversely, TPVs are likely to be longer-lived if they have Arctic, rather than non-Arctic genesis (Fig. 4(e), mean lifetimes 5.2 and 4.4 days, respectively).

Distributions of intensity characteristics are shown in Fig. 4(b,d,f) partitioned into systems with Arctic and non-Arctic genesis. Arctic cyclones with non-Arctic genesis tend be more intense defined either by their maximum $\xi$ or their associated minimum MSLP (at the time of maximum $\xi$). Conversely, TPVs tend to be more intense, diagnosed from their maximum $\theta_{2\mathrm{PVU}}$ anomaly, when their genesis region is in the Arctic (noting though that the number of non-Arctic genesis TPVs is only about a third of that of Arctic genesis TPVs).

A time series of the annual count of tracked TPVs and Arctic cyclones (with lifetimes of at least 1 day) for each year between 1979 and 2018 is shown in the upper panel of Fig. 5, for all systems and split between those of Arctic and non-Arctic genesis. Arctic cyclones are more numerous than TPVs every year, with average annual counts over the 40-year study period of 304 and 208 respectively. TPVs are more likely than Arctic cyclones to have Arctic genesis: 75 % of TPVs have Arctic genesis compared to 55 % for Arctic cyclones. The number of Arctic cyclones (and their Arctic genesis proportion) found here using ERA5 can be compared with the equivalent values found using other reanalyses. Vessey et al. (2020) found an average of 97 (range 96.2–98.3) Arctic cyclones with lifetimes exceeding two days occurred each June–August summer season in the ERA-Interim, JRS-55, MERRA-2 and NCEP-CFSR reanalyses (1980–2017) with an average of 47.4 % (range 47.0–47.8 %) having Arctic genesis. The definitions of the Arctic region and the tracking algorithm and tracked field used in Vessey et al. (2020) are the same as used here; however, in Vessey et al. (2020) T5–T42 spectral filtering was applied to the $\xi_{850}$ field whereas here T5–T63 filtering was used, so retaining some smaller-scale systems. Even adjusting for the longer summer season considered here (five instead of three months), substantially more Arctic cyclones are found although the percentage that have Arctic genesis is similar. The longer minimum lifetime allowed by Vessey et al. (2020) is not the main reason for the disparity as, using the same minimum lifetime as Vessey et al. (2020), we obtain an average of 266 Arctic cyclones. Hence, the higher resolution of ERA5 (compared to the reanalyses used by Vessey et al. (2020)) and allowed smaller scale of tracked cyclones yields the higher cyclone counts, indicating the sensitivity of the results to these factors.

The time series of TPVs and Arctic cyclones are significantly correlated at the 95 % level (Pearson correlation) for both non-Arctic and Arctic genesis (correlation coefficients of 0.366 and 0.324, respectively, with corresponding two-tailed p-values of 0.020 and 0.042, respectively), but this significance drops to the 90 % level (correlation coefficient of 0.299, p-value 0.061) when correlating all Arctic cyclones with all TPVs. The time series do not appear to have notable trends from visual inspection of Fig. 5 and the significant correlations are retained after detrending by linear model fitting (not shown). The number of Arctic cyclones is also correlated with the North Atlantic Oscillation (NAO) during the extended-summer season (index time series downloaded from National Weather Service Climate Prediction Center (2021) and shown in the lower panel of Fig. 5) such that more cyclones occur in the positive phase. The correlations are significant at the 95 % level taking all cyclones and those with non-Arctic genesis only (correlation coefficients of 0.453 and 0.374 and p-values 0.003 and 0.017, respectively). This significance drops to the 90 % level for correlation with Arctic genesis cyclones (correlation coefficient of 0.297, p-

value 0.062). This correlation is consistent with more cyclones tracking northeastwards into the Arctic region from the North Atlantic sector during the positive phase of the NAO when the jet stream is strong. There is no significant correlation between the number of TPVs and the NAO index. The correlation between the time series of Arctic cyclones and TPVs is consistent with TPVs having a role in the initiation and/or intensification of Arctic cyclones, as investigated in the following section.

## 3.2 Spatial association between Arctic cyclones and TPVs

The potential role of TPVs in the initiation and/or intensification of Arctic cyclones is first explored by assessing the proximity between these systems at three independent times (i.e., there is no requirement for sustained association): the times of genesis, maximum growth rate and maximum intensity of the Arctic cyclones. Figure 6 shows these matches for TPVs occurring within 1, 2, 5 and 10° of the Arctic cyclones (termed the overlap radius and defined as the arc length of great circle (geodesic) centred on the Arctic cyclone). As expected, the number of matches increases with the overlap radius. While TPVs are near to some Arctic cyclones at their genesis time, TPVs are more likely to be near to Arctic cyclones when the Arctic cyclones are intensifying rapidly. For the maximum 10° overlap radius considered, 37 % of Arctic cyclones are matched with a TPV at the time of maximum Arctic cyclone growth rate compared to 30 % at the time of Arctic cyclone genesis and this difference is proportionally bigger for 2 and 5° overlap radii (note that TPVs may influence cyclones from varying distances dependent on their structure (see discussion in Sect. 2.2)). This suggests that while TPVs can have a role in the intensification of Arctic cyclones, they are more likely to either co-develop with the Arctic cyclone (i.e. form as a consequence of the three-dimensional baroclinic development) or lead to rapid intensification of an already developing Arctic cyclone as they track into the vicinity of the surface cyclone. The percentage of cyclones matched with TPVs at the time of cyclone maximum intensity is larger than that at the time of maximum growth rate (for a given overlap radius), implying that the spatial association between the TPV is maintained until at least this time. Despite the potential importance of TPVs for Arctic cyclone intensification, the majority (about two thirds) of Arctic cyclones are not within a reasonable upper limit on influence distance (10°) of a TPV at the time of their maximum growth rate.

The geographical locations where the sustained matching between TPVs and Arctic cyclones occurs is now determined using the criteria for matched and unmatched cyclones defined in Sect. 2.2. Maps of the genesis and track densities show large differences between the matched and unmatched cyclones (Fig. 7). The chosen criteria led to a reasonably similar number of cyclones (a few hundred) being identified as matched and unmatched, simplifying the interpretation of the comparison between the density maps. These relatively small cyclone sets arise from the combination of the matching criteria. As for the full set of Arctic cyclones (Fig. 2(a)), large genesis densities of both the matched and unmatched cyclones are found in two localised maxima to the east side of Greenland with larger values in the more northern maxima. Northern Canada, including the Canadian Arctic Archipelago, is a more important genesis region for the matched cyclones. Conversely, the genesis regions over northern Russia and northern Scandinavia (the Arctic Frontal Zone) are relatively more important for the unmatched cyclones. An additional localised maximum over the North Pole also appears for the matched cyclones. As for the full set of Arctic cyclones, the track densities for the matched and unmatched cyclones are much smoother than the genesis densities. The differences between the track densities for the matched and unmatched cyclones can be interpreted using the track densities

for the full set of Arctic cyclones and Arctic genesis TPVs shown in Fig. 3(a) and (e), respectively. As the track density for Arctic genesis TPVs is largest over the Canadian Arctic Archipelago so the track density of the matched Arctic cyclones is also largest there, with an extension westwards over the Arctic Ocean to the north of Canada (to the north of the enhanced genesis density in Fig. 7(a)). In contrast, for the unmatched Arctic cyclones, the track density is largest over the Barents and Kara Seas
along the Russian coastline, consistent with these systems relying on the strongest section of the Arctic Frontal Zone where there is also a low-level jet (Day and Hodges, 2018).

Finally, Fig. 8 shows how the matching changes with month during the extended summer season. The constraint that the cyclones must exist for at least two days prior to their maximum growth rate time has been removed here to generate a larger dataset (of 1226 and 3267 matched and unmatched cyclones, respectively); otherwise the subsets are defined as in Sect. 2.2.
While there is no strong monthly variability, there is a weak tendency for the number of matched cyclones to decrease to a minimum in late summer before increasing again in September with a corresponding increase in the number of unmatched cyclones in late summer. For both the matched and unmatched cyclone sets the count distribution in August has a significantly different arithmetic mean to that in May at the 90 % level according to a two-sided Welch's t-test (without assuming equal variance). However, sequential months are not always significantly different. For comparison, the mean monthly counts of
TPVs with Arctic genesis and Arctic cyclones with maximum intensity in the Arctic with no matching constraints are also shown in Fig. 8. Recall that the constraints for both matched and unmatched cyclones include that the TPVs must have Arctic genesis and that the Arctic cyclones must have maximum intensity in the Arctic. Hence, these constraints have similarly been applied when comparing the monthly variabilities of the matched and unmatched cyclone counts with those of TPVs and Arctic cyclones with no matching constraints. There is little variability in the Arctic cyclone counts. In contrast, the TPV counts have
a minimum in July and so follow an evolution similar to that of the matched cyclone counts. This suggests that the number of TPVs limits the number of matched cyclones in mid-summer with a consequent maximum in the number of unmatched cyclones then. Crawford and Serreze (2015) show that the strength of the Arctic Frontal Zone is maximum in July (see their Fig. 10) and so this may act to maintain Arctic cyclone numbers in mid-summer despite the reduction in TPVs.

### 3.3 Composite structure evolution of Arctic cyclones matched and unmatched with a TPV

The interaction between TPVs and Arctic cyclones is now investigated by comparing the composite structures of the matched and unmatched cyclones (calculated as described in Sect. 2.2). The 200 most intense cyclones from the sets of matched and unmatched cyclones already defined were selected for compositing. The structural evolution of the matched and unmatched cyclones are shown in Figs. 9 and 10, respectively, at four times: two and one days prior to the time of maximum growth rate, maximum growth rate and maximum intensity. The cyclone motion direction (indicated by the grey arrows in Figs. 9 and 10)
is to the right in the composite plots (every cyclone case is rotated so that the orientation of motion vectors align). To simplify the discussion, the right side will be described as "east" in the composite structures. The left columns of each plot show the evolution in near surface fields ($\theta'_{900}$ and MSLP). The anomaly field $\theta'$ (relative to the domain average), is plotted rather than the full $\theta$ field to allow for the seasonal temperature variation over the extended-summer season considered. The evolution of the matched and unmatched composite cyclones are very similar in these fields. A warm sector develops on the southeast flank

of the MSLP centre of the cyclone on a broad northeast–southwest temperature gradient two days prior to maximum growth rate. This reaches a peak $\theta'$ and closest proximity to the centre of the cyclone at the time of maximum growth rate. The cold sector gradually wraps around the cyclone centre to the northwest as the cyclone intensifies and the $\theta'$ at the centre of the cyclone cools between the times of maximum growth rate and maximum intensity. In both composites the minimum MSLP decreases to approximately 990 hPa as the cyclone evolves towards maximum $\xi$ intensity. However, the low-level $\theta$-wave and

MSLP perturbation amplitude are slightly stronger in the unmatched cyclone composite.

The plots in the right columns of Figs. 9 and 10 show, for the same times as the plots in the left columns, vertical cross-sections in the cyclone motion direction through the centres of the composite cyclones. It is assumed that any TPV that affects the evolution of the cyclone lies approximately along these cross-sections, as required for growth by baroclinic instability release. Relative vorticity is shown to indicate the strength and tilt of the cyclonic structure. Selected contours of PV indicate

the dynamical tropopause (the 2-PVU surface) and regions of enhanced lower-tropospheric PV, possibly caused by diabatic processes. Potential temperature contours indicate the change in static stability throughout the troposphere and can be compared to the $\theta'_{900}$ field shown in the plots in the left columns. In contrast to the similarity of the evolution of the near-surface fields for the matched and unmatched cyclones, the vertical cross-sections show distinct characteristics. The composite cyclone evolves from a rearward (westward) tilted structure at the time of maximum growth rate to a more vertical structure at maximum

intensity time in both composites. However, it is striking that unmatched cyclones are dominated by lower-tropospheric $\xi$ at early times and the upper-tropospheric $\xi$ amplifies faster to become comparable to the low-level $\xi$ at time of maximum growth and maximum intensity. This behaviour is consistent with baroclinic wave growth initiated from lower levels, akin to Pettersen type A mid-latitude cyclogenesis, where the tropopause-level anomaly is generated as a result of the baroclinic interaction with low levels. The region of enhanced $\xi$ also extends further northwards with height in both composites at the time of maximum

growth rate, although the northwards distance of the peak $\xi$ is similar (not shown). In the cyclones matched with TPVs, the $\xi$ at tropopause level is approximately equal to the low-level $\xi$ at the earliest time shown and it amplifies faster in the composites such that it dominates at the time of maximum growth rate. At maximum intensity, the cyclone core is stronger throughout the depth of the cyclone for the matched composite. Consistent with the stronger cyclonic $\xi$ at the tropopause in the matched cyclone composite, the tropopause extends down to higher pressures, reaching close to 550 hPa at maximum intensity time.

Thermal wind balance relates the vertical gradient in $\xi$ to the horizontal curvature in the $\theta$-anomaly field through the following equation:

$$\frac{\partial \xi}{\partial z} = \frac{g}{\theta_0} \nabla^2 \theta',$$

where $g$ is the acceleration due to gravity and $\theta_0$ is a reference $\theta$ value. At maximum intensity the upper- and lower-tropospheric features align to form a columnar vortex in which $\xi$ increases with height, $\nabla^2 \theta' > 0$ and therefore a minimum in $\theta'$ is expected

on all pressure levels. In other words, the matched cases evolve to a cold-core vortex ($\theta$-surfaces bow upwards) throughout the troposphere because their $\xi$ is dominated by the TPV at upper levels. The $\theta$-signature in the lower-troposphere is weak in the composite, but can be strong in individual cases. Enhanced PV values also extend throughout the troposphere at time of maximum intensity in both composites, but with larger values in the matched composite. Note that at one day before the

maximum growth, at 650 hPa the PV tilts eastwards with height (opposite tilt to $\xi$). This is expected for baroclinic growth over height levels where the low-level thermal anomaly dominates the wind field in PV inversion (Methven et al., 2005). It arises because the meridional gradient of low-level $\theta$ is negative (as can be seen in the composite maps) while the background PV gradient is positive, so northward displacement of air is associated with lower boundary $\theta' > 0$ (inducing cyclonic flow) but a negative PV anomaly (inducing anticyclonic flow) so that the net vorticity anomaly has the opposite sign to the PV anomaly near the lower boundary. By the time of maximum growth rate, the upper-tropospheric feature is dominant and large amplitude in the sense that the tropopause comes down low within the TPV, particularly in the matched composite. In this situation, the PV also tilts westwards with height. In both the matched and unmatched composites, there is a secondary maximum in PV near the ground. These features indicate the influence of diabatic and frictional processes since the development of a new maximum could not occur in conservative flow. Attribution of the mid-tropospheric PV anomaly in Arctic cyclones to individual processes is beyond the scope of this paper but could be investigated for individual cases using methods previously used for extratropical cyclones such as "PV tracers" (e.g, Chagnon et al., 2013; Stoelinga, 1996) or Lagrangian trajectories (e.g., Joos and Wernli, 2012).

The evolution of the composite structure tilt in $\xi$, relative to the cyclone motion direction (calculated as described in Sect. 2.2), is summarised for the matched and unmatched cyclones in Fig. 11. The tilt upshear (i.e., rearwards relative to the cyclone propagation direction) is greatest at time of maximum growth rate, the necessary configuration for baroclinic growth. The tilt is also rearwards at earlier times in both composites, although more strongly in the unmatched cases. As baroclinic waves grow the tilt is expected to tend towards a phase-locked configuration between tropopause-level and low-level counter-propagating Rossby waves—if the tilt is initially less than this then it will increase with time as the wave amplifies which is counter to the differential advection of the disturbances by the shear flow (Heifetz et al., 2004). This may explain the composite behaviour observed. However, it is also possible that the structures of systems averaged in the composite are most similar in their phase-locked configuration at time of maximum growth, and the weaker tilt and amplitude at earlier times is a result of averaging together systems with different structures. After the time of maximum growth, the tilt reduces markedly. However, despite this general similarity in evolution, there are differences between the two composites that are robust given the marked standard errors in the $\xi$ centre displacements. The matched cyclones acquire a maximum tilt distance (distance between the upper-tropospheric maximum in $\xi$ and that at 900 hPa) of $\approx 150$ km compared to $\approx 180$ km for the unmatched cyclones at the time of maximum growth rate. The unmatched cyclones are more strongly tilted than the matched cyclones in the hours up to and including the time of maximum growth rate. These findings are consistent with the matched Arctic cyclones interacting with an isolated TPV shortly before the time of maximum growth rate and forming a single columnar vortex structure, while unmatched cyclones have rearward tilts at all stages, consistent with baroclinic wave growth through interaction between the low-level wave on the Arctic Frontal Zone and a tropopause-level disturbance.

## 3.4 Comparison of Arctic and mid-latitude cyclone structure

The structural evolution of these Arctic summer cyclones, as shown in Figs. 9–11, can be compared with that of mid-latitude winter cyclones, beginning with the near-surface structure. Dacre et al. (2012) generated a cyclone atlas by compositing the

200 most intense winter (December–February) North Atlantic cyclones from 1989 to 2009 using ERA-Interim data. Although a warm sector develops with a similar structure in the Arctic cyclones and to that found for the North Atlantic cyclones, the relatively cold core of the Arctic cyclones contrasts with the warm core typical of mid-latitude cyclones (see frontal locations in Fig. 3 of Dacre et al. (2012) and also low-level temperature composites generated using the associated cyclone atlas website). A relatively warm core at maturity is also consistent with the two predominant conceptual models of mid-latitude cyclones: the Norwegian cyclone model with occluded front wrapping to the north of the cyclone centre and the Shapiro-Keyser model with a warm air seclusion (Shapiro and Keyser, 1990). These summer Arctic cyclones are substantially weaker in terms of MSLP at their maximum intensity time than the winter mid-latitude cyclones (minimum MSLP of $\approx 990$ hPa compared to $< 970$ hPa in the cyclone atlas). However, summer mid-latitude cyclones are also weaker than winter cyclones. For example, Čampa and Wernli (2012) found that the modal minimum sea level pressure for all tracked winter cyclones (also using ERA-Interim data) was 970–990 hPa for cyclones in the regions of the Gulf of Alaska and between Greenland and Iceland in winter, but 990–1010 Pa in summer.

The structural evolution of the $\xi$ cross-sections in the Arctic cyclone composites can also be compared to equivalent composites for winter mid-latitude cyclones generated using the cyclone atlas website which show the largest values of $\xi$ are present at near-surface levels one and two days prior to time of maximum intensity. This structure thus more closely resembles that for the unmatched than the matched cyclones. The vertically-coherent PV feature found in both cyclone composites was termed a "PV tower" by Rossa et al. (2000) and is characteristic of mature mid-latitude cyclones (Čampa and Wernli, 2012). Rossa et al. (2000) showed that the PV in the tower in a late autumn case study had three distinct origins: the top portion was stratospheric air in the region of lowered tropopause; the bottom portion was PV generated in the boundary layer by non-conservative diabatic processes (both friction and heating); and the middle portion was also generated by diabatic processes (chiefly latent heating in ascending air). Such PV towers can also occur in strong summer mid-latitude cyclones and Martínez-Alvarado et al. (2016) analysed strong diabatic contributions to the tower in such an event. Vertically-coherent structure has also been found in Arctic cyclones; for example, Tao et al. (2017b) discuss the origins of the equivalent barotropic structure (from the surface to the lower stratosphere) found in their case study. However, in these Arctic composites the middle PV anomaly is much weaker than for mid-latitude cyclones, suggesting that latent heat release is less important to this structure than in mid-latitude cyclones. The tropopause-level structure in the matched cyclone composite can be compared to that of composite TPVs. Figure 9 of Cavallo and Hakim (2010) shows composite cyclonic TPV structures produced using 568 TPV samples simulated using the Weather Research and Forecasting (WRF) model with 30-km horizontal grid spacing and 31 vertical levels. The tropopause fold extends down to about 550 hPa, similar to the matched cyclone composite at maximum intensity time. A cold anomaly is present at and below the tropopause in both the TPV composite and matched cyclone composite. Additionally, the peak in the meridional wind speed at tropopause level in the TPV composite is consistent with the peak in $\xi$ at tropopause level in the matched composite.

Finally, the evolution of vorticity tilt in the Arctic cyclones can be compared that of mid-latitude cyclones. Figure 7 of Bengtsson et al. (2009) shows, calculated using the same methodology as used here, the tilt evolution of the 100 most intense cyclones found in the 40-year ECMWF reanalysis (ERA-40) and a 32-year, end of twentieth century, climate integration with

the ECHAM5 model (though note that in Bengtsson et al. (2009) the 36 hours on either side of the maximum intensity time rather than the 48 hours on either side of the time of maximum growth rate are considered). The general increase in upstream tilt as the cyclones intensify, followed by a decrease to near vertical orientation at their time of maximum intensity is a trait shared by the mid-latitude and Arctic cyclone composites. However, in their mid-latitude composite, the maximum upstream tilt was $\approx 3-4°$ great circle separation ($\approx 330-440$ km) in $\xi$ centres at 900 and 200 hPa, roughly coincident with the maximum growth rate time. Although this separation is about double that found in our unmatched Arctic cyclone composite, Arctic cyclones are typically much smaller scale. One reason is that the Rossby radius is at most 500 km for the Arctic cyclone environment, approximately half the value typical of mid-latitude systems. Some of this difference may also be attributable to the smaller, and so more extreme in intensity, sample used in Bengtsson et al. (2009) (the 100 rather than 200 most intense cyclones). However, this effect is likely to be somewhat compensated for by the compositing here relative to the time of maximum growth rate (when the tilt should be strongest) rather than time of maximum intensity (meaning that the cyclones contributing to Fig. 7 of Bengtsson et al. (2009) are unlikely to all have their maximum tilt at the same time). Nevertheless, for the Arctic cyclones matched with TPVs the upshear tilt is still smaller than in mid-latitude cases, even as a proportion of cyclone scale.

## 4  Conclusions

Arctic cyclones are the major weather-related hazard in the Arctic. However, in comparison with mid-latitude cyclones there have been very few studies examining their structure, evolution and mechanisms for growth. Here the focus has been on summer-time Arctic cyclones because human activity in the Arctic is greatest in this season and because they play a major role in modifying the sea ice distribution over the marginal ice zone where ice fraction is less than one. Arctic cyclones in summer also typically have much larger scale than the intense polar lows that occur in winter. Case studies of Arctic cyclones in summer have focused on the most intense or long-lived examples. Most studies (see Sect. 1) refer to interaction with a tropopause polar vortex (TPV) disturbance and imply that such disturbances are important to the growth of the Arctic cyclone. However, this link has not been explored taking into account a more complete climatological set of Arctic cyclone and TPV events.

The chief purpose of this paper is to characterise and quantify the proportion of Arctic cyclones that are near to a TPV (with genesis in the Arctic) at the time of maximum cyclone growth rate through to maximum intensity (termed matched cyclones), and to examine the average structure in matched and unmatched cases using a statistical composite approach. To the authors' knowledge, this is the first paper in which both low-level Arctic cyclones and TPVs have been tracked using the same tracking algorithm and reanalysis dataset (and the first use of ERA5 for such tracking), enabling statistics of genesis and tracks and to be compared systematically. Furthermore, the ERA5 data have been used to create dynamically consistent composites of the three-dimensional Arctic cyclone structures from the surface to the tropopause region enabling quantification of system tilt and deductions of the nature of the growth mechanisms.

The first research question (defined in Sect. 1) was to determine the characteristics of TPVs and how they compare to those of Arctic cyclones. The locations of enhanced genesis and track densities of Arctic cyclones and TPVs were shown to be consistent with previous climatologies produced independently using other datasets (Cavallo and Hakim (2009) and Vessey

et al. (2020), respectively). The frequency distributions of lifetimes and intensities were then calculated for the tracked Arctic cyclones and TPVs and also split into those with genesis within and outside the Arctic. While the modal lifetime of Arctic cyclones of 2–3 days exceeds that of TPVs, TPVs have a longer mean lifetime (5.0 compared to 4.4 days) skewed by the much longer tail of the TPV lifetime distribution (the longest TPV track was 43 days). The lifetime and intensity characteristics also depend on the genesis location of the features. Cyclones that track into the Arctic have longer mean lifetimes and are also more intense on average than those that have their genesis within the Arctic. In contrast, TPVs with genesis within the Arctic are much more frequent and more intense on average than those with genesis further south. The annual counts of Arctic cyclones and TPVs are significantly correlated at the 95 % level for features with both Arctic genesis and non-Arctic genesis suggesting that TPVs have a role in the initiation and/or intensification of Arctic cyclones.

The second research question addressed the role of TPVs in the initiation and intensification of Arctic cyclones. The percentage of Arctic cyclones instantaneously associated with a TPV (varying the great circle separation criterion in the range $1–10°$) increases from 1–30 % at the genesis time of Arctic cyclones to 1–37 % at the time of maximum growth rate and 3–42 % at the time of maximum intensity. Hence, even with the greatest separation criterion (approximately twice the Rossby radius) the majority of Arctic cyclones (about two thirds) are developing without a TPV in close proximity. Also these statistics suggest that TPVs do not play a part in the genesis of many Arctic cyclones, but have a potential role in subsequent Arctic cyclone growth. The geographical distribution of matched cyclone cases, with sustained association with a TPV during their intensification, and unmatched cyclone cases, with no interaction with a TPV, differs markedly. Matched cyclones preferentially track over the Arctic Ocean to the north of the Alaskan and Canadian coastline, and the Canadian Archipelago. In contrast, unmatched cyclones track preferentially over the Barents and Kara Seas to the north of the Russian coastline. TPV track density is higher across the North American Arctic coastline and Canadian Archipelago region, accounting for the higher proportion of matched Arctic cyclone cases in this area. The unmatched cyclone cases have a higher genesis frequency over northern Eurasia, both within the Arctic region and south to approximately 55°N, associated with the strong low level baroclinicity, especially along the Arctic Frontal Zone associated with the Arctic Ocean coastline in summer. There is a weak tendency for the number of matched cyclones to decrease to a minimum in August of the extended May–September summer season with a corresponding increase in the number of unmatched cyclones.

Finally, the third research question addressed how the evolution of Arctic cyclone structure and intensity is modified by interaction with TPVs. The matched and unmatched cyclones were compared by compositing the 200 most intense cyclones from each cyclone set at four specific times, tracing back the structural evolution: the time of Arctic cyclone maximum intensity, maximum growth rate, then one and two days prior to maximum growth rate. Note that the rationale for using twice the Rossby radius as the distance criterion distinguishing matched and unmatched cases is that the Rossby radius characterises the range of the velocity field induced by a mesoscale PV structure such as a TPV (with radius $r < L_R$). However, TPVs can be larger than this, or be embedded in a large-scale trough, when their far field influence would extend to greater distances. Therefore, this distance criterion is not conservative in that it allows for TPVs of varying structures, and orientations relative to the surface cyclone, to be considered within range for baroclinic interaction in all cases identified as matched. Some cases with potential for interaction may not be identified as matched, but, importantly, are also unlikely to be identified as unmatched due to the

different criteria for that category and such cases do not contribute to the composite structures shown. The two composite sets revealed distinct structural evolution and the three-dimensional structure of dynamical fields was used to deduce the relative importance of mechanisms acting in each set.

The unmatched Arctic cyclones have a larger amplitude low-level $\theta$-wave on average and develop a more pronounced warm sector by the time of maximum growth rate. The $\xi$ is dominated by low levels two days prior to maximum growth rate and then upper-tropospheric $\xi$ grows faster so that it attains a similar magnitude near the tropopause and surface with a pronounced upshear tilt with height (i.e., rearwards relative to the cyclone motion vector) which is strongest at time of maximum growth rate. There is no pre-existing (tracked) tropopause disturbance and the anomaly here grows in these two days as a result of the mutual growth with the lower wave. These structural features are all commensurate with the baroclinic growth mechanism dominated by a low-level disturbance on a baroclinic zone at initial time: type A cyclogenesis. Between the time of maximum growth rate and maximum intensity the tilt reduces, but remains upshear on average. Finally, the track density map for unmatched cases shows how they preferentially track along the north coast of Russia, but the genesis can be further south, associated with crests of waves on the strong baroclinic zone there.

In contrast, the matched Arctic cyclones are upper-troposphere dominated in the upshear-tilted baroclinic configuration at the time of maximum growth rate. This upper-tropospheric vorticity disturbance is identified with the tracked TPV in these cases. The tropopause is markedly lower in association with the upper PV anomaly in this composite (down to about 550 hPa compared to about 475 hPa for the unmatched composite). Interestingly, two days before maximum growth rate the upper and lower $\xi$ are equally weighted in the composite, and by one day before maximum growth rate the $\xi$ tilts slightly westwards and the PV tilts eastwards in the lower troposphere. These are all features consistent with a dry baroclinic growth mechanism through cooperative interaction between an tropopause-level PV disturbance and lower-boundary $\theta$-wave. However, the upper disturbance is so dominant by the time of maximum growth that the PV tilts westwards at all levels. By the time of maximum intensity, there is some enhancement of mid-tropospheric PV as well as the TPV and boundary layer PV anomalies in both composites, indicating some influence of latent heat release. However, latent heat release is not as important in this type of cyclone compared with strong mid-latitude cases because a strong PV tower does not develop.

The matched cyclone composite has some features that differ markedly from mid-latitude composite structures. The upshear tilt is relatively weak, even at the time of maximum growth rate. There is no detectable tilt on average at the time of maximum intensity: the structure forms a single columnar vortex. The $\xi$ is upper-tropospheric dominated and therefore thermal wind balance implies that it must have a cold core and $\theta$-surfaces bow upwards in the centre of the cyclone. The structure is very similar to the wind and thermal structure obtained by inversion of an isolated tropopause-level PV anomaly using a balance approximation (e.g., see Fig. 1 in Thorpe, 1986) and characteristic of the structure of isolated TPVs (Cavallo and Hakim, 2010). Near the lower boundary, a warm sector forms to the southeast of the cyclone centre (assuming eastwards cyclone motion), while the cold air wraps around the composite cyclone centre to the northwest, but the dominance of the upper-level PV results in a cold core surface at maturity. This contrasts with the composite structure of intense mid-latitude cyclones which tend to form a warm seclusion at low levels in their mature phase and are therefore warm core in the lower troposphere, even though they are cold core in the upper troposphere. For example, Martínez-Alvarado et al. (2014) show a dropsonde curtain

from the centre of an intense cyclone of the Shapiro-Keyser type through the bent-back warm front that arcs round the warm seclusion. The front slopes radially outwards with height where $\theta$-surfaces dip steeply downwards towards the cyclone centre in the lower troposphere.

One plausible explanation for the dynamical behaviour in the matched Arctic cyclone cases is that they develop away from a strong baroclinic zone and associated jet stream. The final stages of growth arise from the baroclinic interaction of the pre-existing TPV, with its strong PV anomaly on the tropopause, with a low-level cyclone that has already formed. Theoretical studies based on quasi-geostrophic dynamics have shown the tendency for upper and lower vortices to approach and align one above the other (Polvani, 1991) or for isolated tilted vortex disturbances to become upright (Reasor and Montgomery, 2001). Such behaviour is prohibited if the large-scale vertical shear is too strong.

In summary, this research has demonstrated that TPVs are instantaneously in close proximity (within twice the Rossby radius) of Arctic cyclones at their time of maximum growth rate in about one third of cases. TPVs are less likely to be within this range at the initial cyclogenesis. However, it is found that more Arctic cyclones do tend to occur in summers with increased numbers of TPVs. The track density pattern of Arctic cyclones matched with TPVs (and so having a sustained association with TPVs during their intensification) is geographically distinct from that of unmatched cyclones: matched Arctic cyclones preferentially track along the North American shore of the Arctic Ocean and the Canadian Archipelago, whereas unmatched cyclone tracks are much more frequent along the Eurasian shore of the Arctic Ocean. The cyclones in matched cases are dominated by the flow associated with the upper-level PV and develop from upper-level precursors, while the cyclones in the unmatched cases grow from low-level warm anomalies extending polewards on the Arctic Frontal Zone and the upper-level PV anomaly grows rapidly as a result of advection by the winds associated with the low-level disturbance during baroclinic growth. Therefore, the unmatched cases have more similarity with mid-latitude cyclone dynamics, while the Arctic cyclones matched with TPVs more closely resemble the multi-layer interaction between isolated vortices.

*Code and data availability.* ERA5 data are freely available and downloadable from the Copernicus Climate Data Store (https://cds.climate.copernicus.eu). The TRACK algorithm is available on the University of Reading's Git repository (GitLab) at https://gitlab.act.reading.ac.uk/track/track.

*Author contributions.* SLG and JM designed the study. KIH performed the feature identification, tracking and matching and provided some plotting code. JLV performed preliminary analysis as part of his M.Sc. dissertation supervised by the other authors and further analysis was performed by the other authors after the dissertation was completed. SLG drafted the paper and produced the final plots. The other authors provided edits to the paper.

*Competing interests.* No competing interests are present

*Acknowledgements.* The authors acknowledge the European Centre for Medium Range Forecasting (ECMWF) for the production of ERA5 dataset. The results contain modified Copernicus Climate Change Service information [1979-2018]. Hersbach et al. (2018b) and Hersbach et al. (2018a) were downloaded from the Copernicus Climate Change Service (C3S) Climate Data Store. Neither the European Commission nor ECMWF is responsible for any use that may be made of the Copernicus information or data it contains. KIH's contribution was funded by the United Kingdom's Natural Environment Research Council (NERC) as part of the National Centre for Atmospheric Sciences.

670

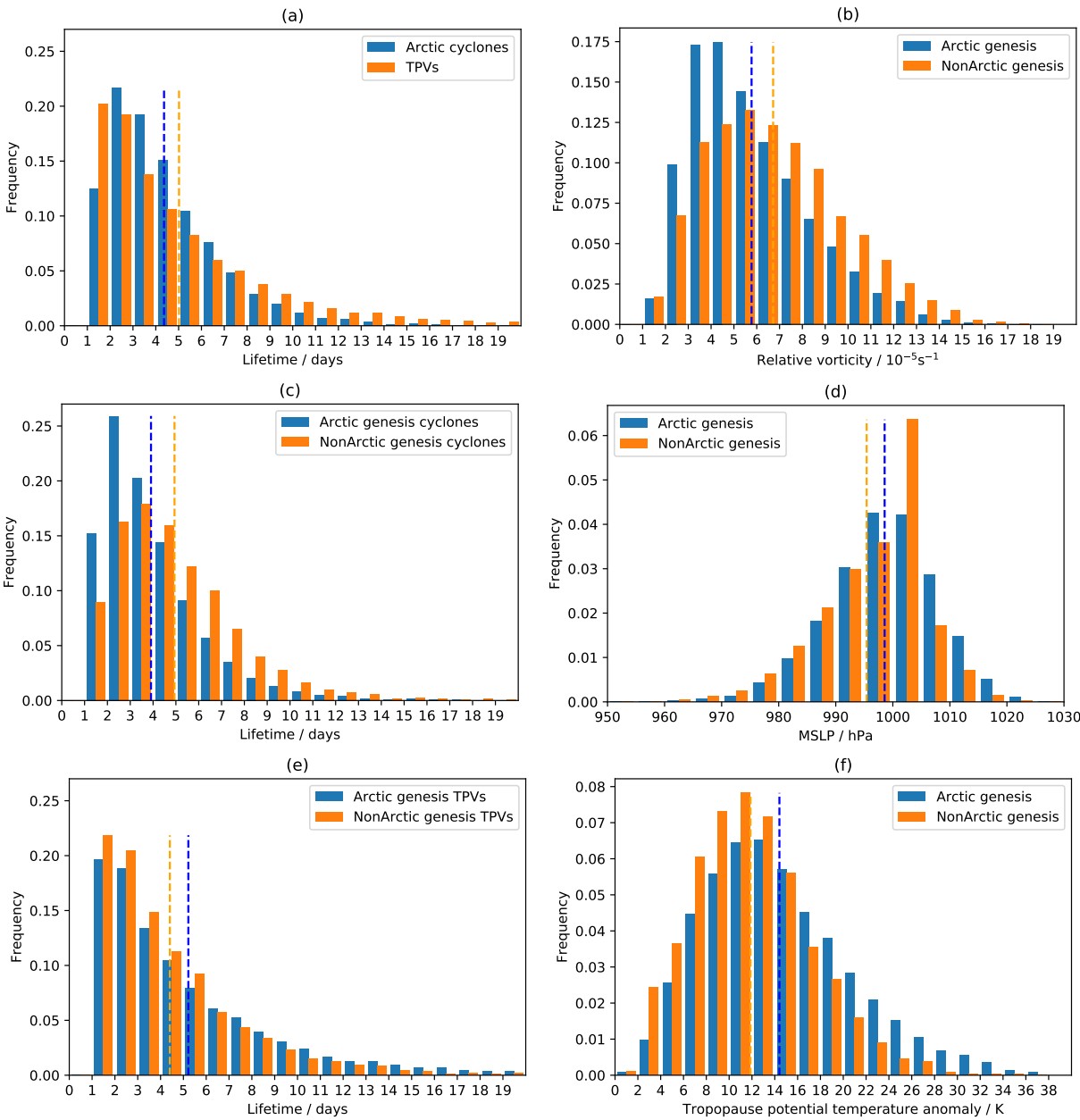

**Figure 4.** Normalised frequency plots of Arctic cyclone and TPV characteristics: (a,c,e) lifetimes of Arctic cyclones and TPVs out to 20 days (maximum lifetimes are 27 and 43 days respectively) with (a) for all Arctic cyclones and TPV systems, (c) for Arctic cyclones partitioned into Arctic and non-Arctic genesis, and (e) for TPVs partitioned into Arctic and non-Arctic genesis; and (b,d,f) intensity of systems partitioned into Arctic and non-Arctic genesis with (b) maximum filtered $\xi_{850}$ of Arctic cyclones, (d) MSLP at the time of maximum $\xi_{850}$ of Arctic cyclones, and (f) maximum filtered $\theta'$ on the 2-PVU surface. Mean values are indicated by vertical dashed lines in each panel. The minimum lifetime of tracked systems is 1 day. In (a,c,e) the first pair of bars presents the frequency of systems with 1 day$\leq$lifetime$<$ 2 days, the second pair systems with 2 days$\leq$lifetime$<$ 3 days etc. An analogous interpretation of the bars applies for the intensity plots.

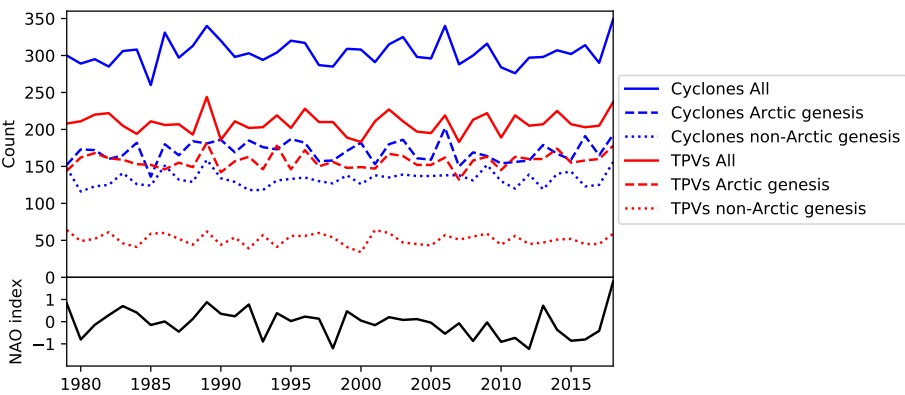

**Figure 5.** Time series of extended-summer counts of Arctic cyclones and TPVs with a lifetime of ≥1 day (upper panel) and the average extended-summer NAO index from 1979–2018 (lower panel).

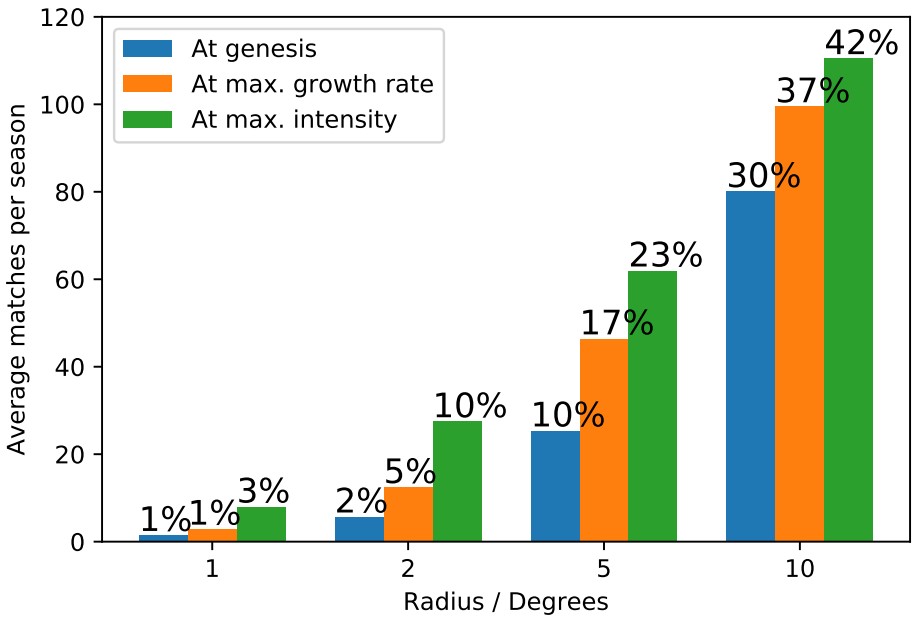

**Figure 6.** Matches between (all) Arctic cyclones and Arctic-genesis TPVs, with lifetimes ≥ 2 days, independently at times of genesis, maximum growth rate and maximum intensity of the Arctic cyclone for selected spatial overlap radii (note that 1 degree is equivalent to 111 km in great circle distance). Values above the bars are the percentage of Arctic cyclones that are matched (of the 10636 Arctic cyclones with lifetimes ≥ 2 days).

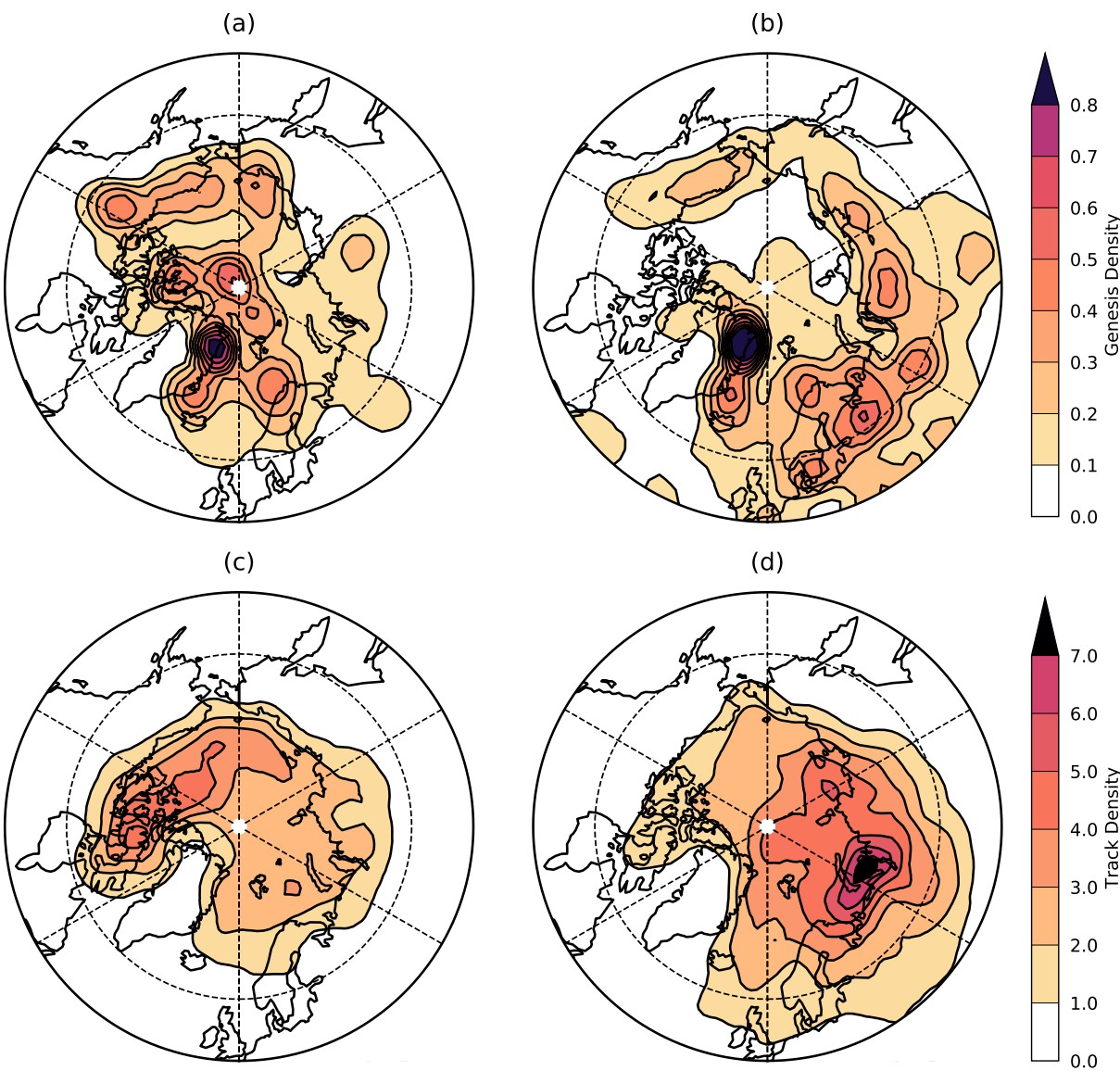

**Figure 7.** Genesis and track densities of Arctic cyclones matched and unmatched to a TPV. (a) matched and (b) unmatched cyclone genesis densities and (c) matched and (d) unmatched cyclone track densities during the extended summer season between 1979 and 2018. Units are number per unit area per season where the unit area is equivalent to a 5 degree spherical cap ($\sim 10^6$ km$^2$). The total number of tracks is 302 for the matched cyclones and 431 for the unmatched cyclones. Maps are orientated with $0°$ longitude at the bottom.

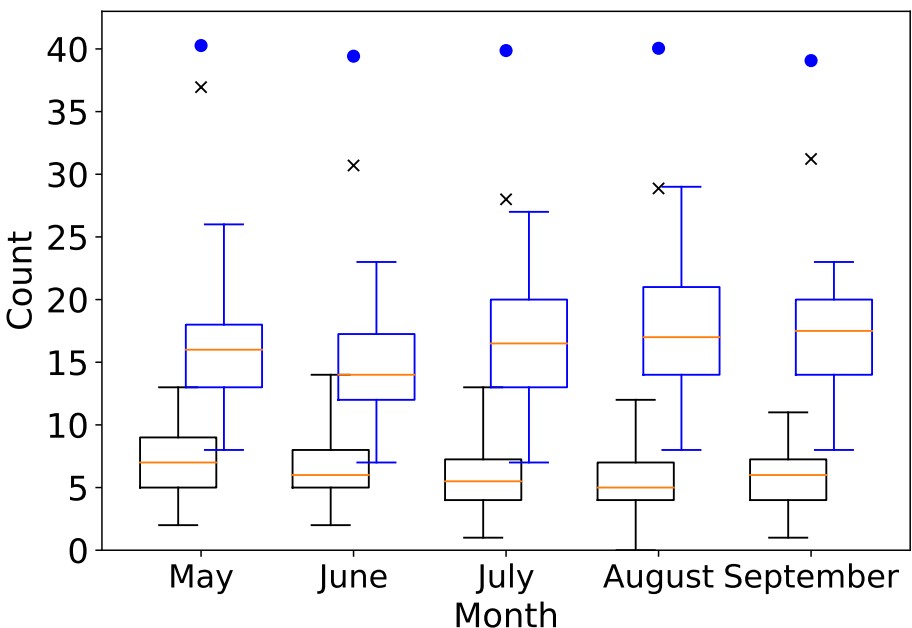

**Figure 8.** Box and whisker plot of the monthly counts of the matched (black) and unmatched (blue) cyclones. The median values for the 40-year dataset are given by the orange bars in the boxes and the boxes span the interquartile range. The whiskers extend to show the full range of the data. The box and whiskers for the matched and unmatched cyclones are slightly offset for clarity. Also shown are the mean monthly numbers of TPVs with Arctic genesis (black crosses) and Arctic cyclones with maximum intensity in the Arctic (blue circles). All data are for systems with lifetimes $\geq 1$ day.

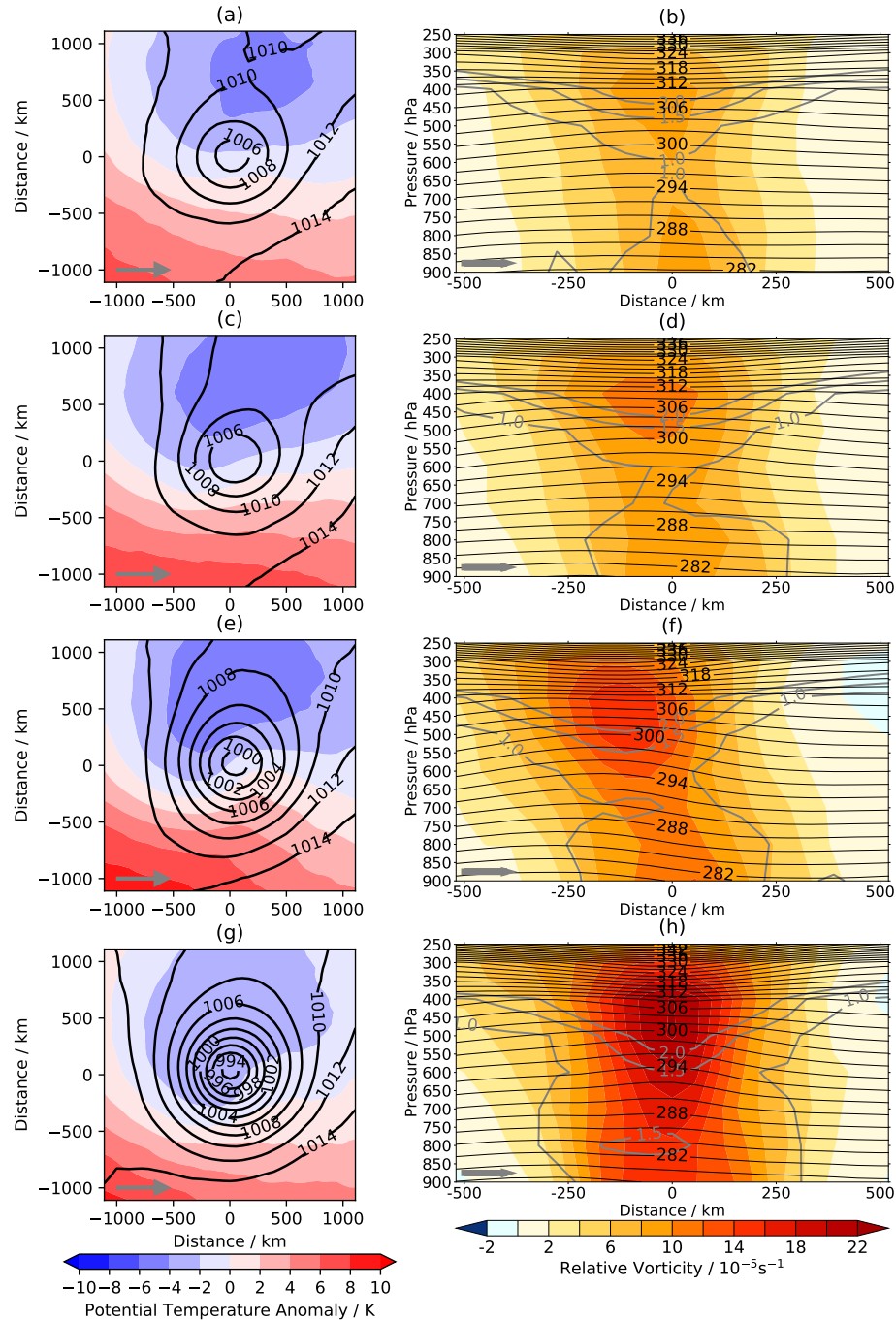

**Figure 9.** Composite fields of the 200 most intense matched Arctic cyclones at times (a,b) two days prior to maximum growth rate, (c,d) one day prior to maximum growth rate, (e,f) maximum growth rate, (g,h) maximum intensity. Left column shows the horizontal distributions of $\theta_{900}$ anomalies relative to the domain average (shaded) and MSLP (hPa). Right column shows vertical cross-sections of $\xi$ in the along-track direction (shaded). Thick, grey contours: potential vorticity (1, 1.5 and 2 PVU shown). Black contours: $\theta$ (interval 2K). Grey arrow in each panel indicates direction of cyclone motion. The composites use full resolution (rather than filtered) fields.

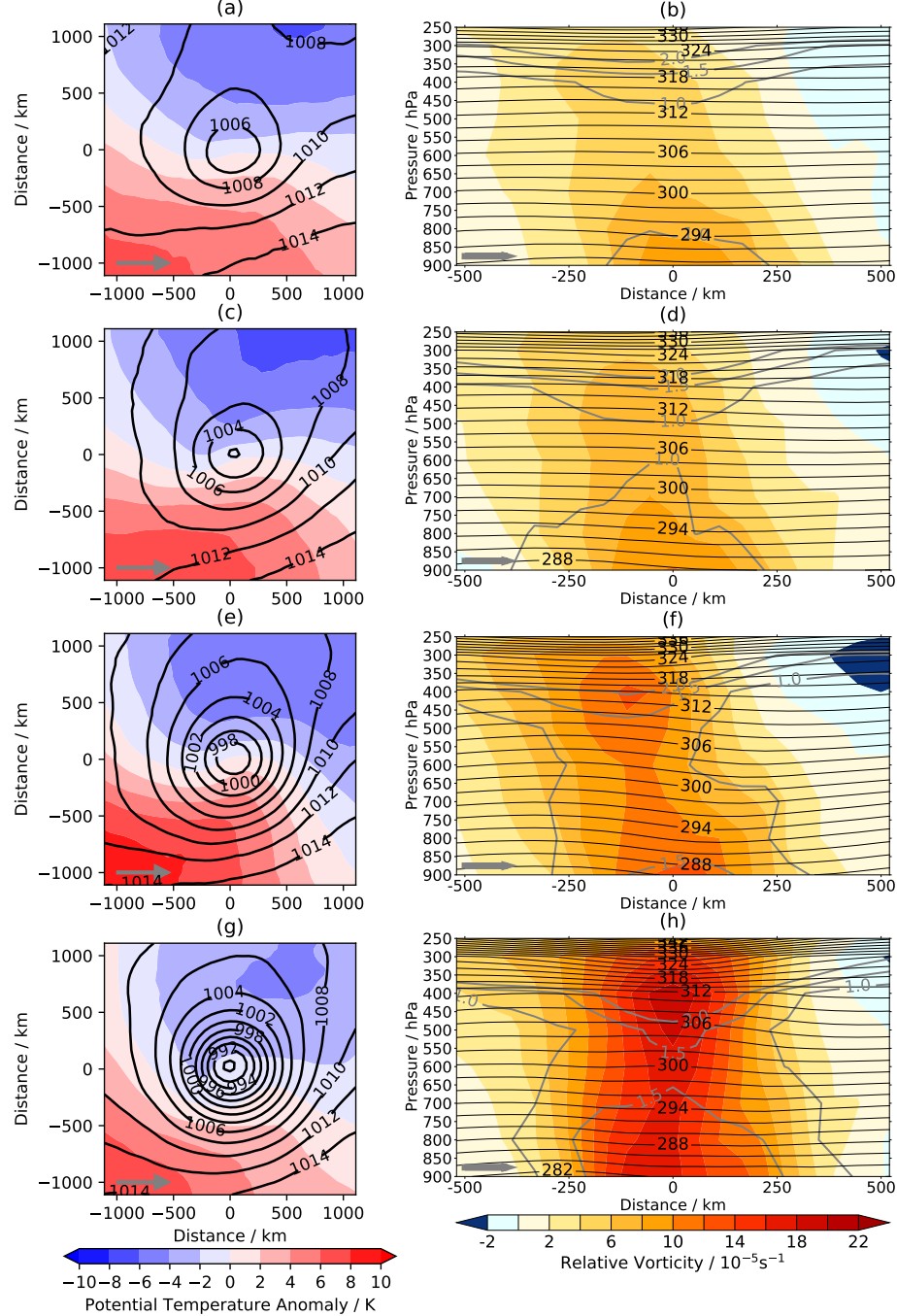

**Figure 10.** As for Fig. 9 but for the 200 most intense unmatched Arctic cyclones.

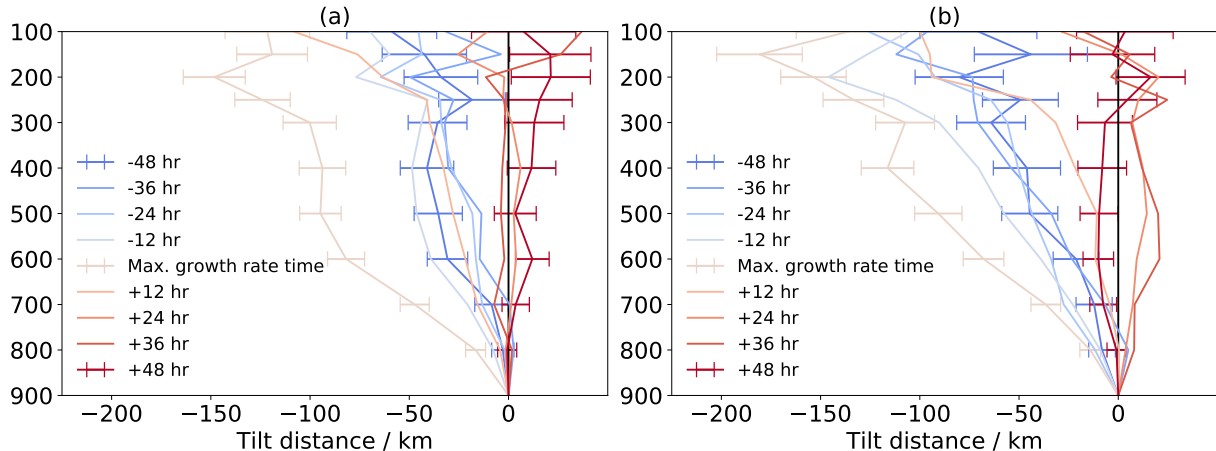

**Figure 11.** Evolution of composite vorticity tilt structure from 48 hours before to 48 hours after the time of maximum growth rate for the 200 most intense (a) matched and (b) unmatched cyclones. Illustrative standard error bars are shown at three times. Tilt distance is the great circle distance relative to the $\xi$ centre at 900 hPa

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
