# Peer review of "The role of tropopause polar vortices in the intensification of summer Arctic cyclones"

_Weather and Climate Dynamics, 2021_

## Author Response (AR1)

**The role of tropopause polar vortices in the intensification of summer Arctic cyclones by Gray, Hodges, Vautrey and Methven: Response to**

**reviewers**

The reviewers' comments are copied below in black with our point-by-point responses in blue.

**Response to reviewer 1:**

This submission undertakes a comprehensive analysis (based on 40 years of ERA5 reanalysis) of summer Arctic TPV influences on the intensification of near-surface cyclones. As the authors point out, there has been relatively little attention devoted this topic. The ideas, methods etc. used in the paper are described well and clearly, and the rationale and aims are succinctly expressed. In part, the investigation includes a careful stratification of the upper and low-level systems which serves to reveal much about the connections between these, and the accompanying analysis reveals some very interesting (and perhaps unexpected) results.

The submission is poised to make a significant contribution to the literature, but not in its present form. Before I would be able to recommend acceptance, there are a number of issues which need to be addressed.

We thank the reviewer for taking the time to review our paper and for his/her comments on the importance of our research.

Lines 29-31: In this connection with the AFZ very valuable to reference the works of

Crawford, A., and M. Serreze, 2015: A new look at the summer Arctic Frontal Zone. Journal of Climate, 28, 737-754, doi: 10.1175/jcli-d-14-00447.1.

Crawford, A. D., and M. C. Serreze, 2016: Does the summer Arctic frontal zone influence Arctic Ocean cyclone activity? Journal of Climate, 29, 4977-4993, doi: 10.1175/JCLI-D-15-0755.1.

Crawford, A. D., and M. C. Serreze, 2017: Projected changes in the Arctic Frontal Zone and summer Arctic cyclone activity in the CESM Large Ensemble. Journal of Climate, 30, 9847-9869, doi: 10.1175/JCLI-D-17-0296.1.

Thank you for highlighting these relevant papers. We now cite the second of these (Crawford and Serreze, 2016) in our paper in relation to our brief introduction to the Arctic frontal zone. Citation of the Crawford and Serreze (2017) paper would have required additionally citing several other papers that discuss the changes in projected summertime Arctic cyclone activity linked to the Arctic frontal zone. As our paper is not about either the impact of climate change on Arctic cyclones, or the impact of the Arctic frontal zone on cyclone activity, a detailed review of this literature would be excessive and disrupt the flow of this early part of the introduction.

Crawford and Serreze (2015) is also now cited in the last paragraph of section 3.2.

Line 37: 'Cavallo et al., 2009' should be Cavallo and Hakim, 2009'. (Similar error at lines 66, 104, 244, 261, 495, ... and maybe elsewhere.)

Thank you for pointing this out. The error came from duplicate author names being included in the bibtex file and this has now been fixed.

Lines 41-65: A informative survey is presented here in connection with Arctic cyclone numbers and trends, and how results may depend on a range of techniques used. Beneficial here to cite analysis

of Screen and coauthors, 2018: Polar climate change as manifest in atmospheric circulation. Current Clim. Change Reports, 4, 383-395, and Rudeva et al., 2015: Variability and trends of global atmospheric frontal activity and links with large-scale modes of variability. Jnl. Clim., 28, 3311-3330.

Thank you for highlighting these relevant papers. The Screen at al. (2018) paper is a review article that also cites several other papers in addition to the Rudeva and Simmonds (2015) paper on this topic. Hence, we have just cited the Screen et al. paper.

Lines 73-78: This text presents a nice clarification of many terms which, in this broad topic, have often been confused. Incidentally, in connection with 'radiative cooling re-building the PV reservoir' it would be helpful to reference the study of

Cavallo, S. M., and G. J. Hakim, 2012: Radiative impact on tropopause polar vortices over the Arctic. Monthly Weather Review, 140, 1683-1702, doi: 10.1175/MWR-D-11-00182.1.

Thank you for the suggestion. Cavallo and Hakim (2012) examine the importance of radiative cooling for the formation and characteristics of TPVs. Our sentence "Eventually, the larger-scale tropospheric polar vortex is re-established in autumn as a result of increasing net radiative cooling re-building the PV reservoir." refers to radiative cooling across the Arctic and is not specific to radiative cooling within TPVs. Hence, the addition of this citation is not appropriate here.

Line 93: Change 'Cavallo et al. (2010)' to 'Cavallo and Hakim (2010)' – similar change required at II. 119, 177, 453, 552, ...

Thanks for pointing this out. The error came from duplicate author names being included in the bibtex file and this has now been fixed.

Line 127: See also the detailed vorticity budget analysis of this storm by

Ryota Ishiyama, and Hiroshi L. Tanaka, 2021: Analysis of vorticity budget for a developing extraordinary Arctic cyclone in August 2016. Sola, doi: 10.2151/sola.2021-020.

Thank you for pointing out this newly published paper, a citation has been added to the text.

Line 146: I suggest starting a new paragraph with the 'In this study ...'. This makes it clear that the introductory survey has finished, and the authors are now going on to detail what they plan to achieve in the paper.

We thank the reviewer for this suggestion. We prefer to not to break the paragraph here though as the sentence beginning "In this study we explore this research gap..." links directly to the preceding text in the paragraph which states what this research gap is. If we instead started a new paragraph with this sentence, we would need to restate the research gap.

Line 352-354: It is not clear whether we are looking at statistically significant differences between the months. This should be checked. If, e.g., an F test does not indicate significance perhaps this comment should be deleted, as we would just be looking at noise. If the differences are above the noise level, some (thermo)dynamic explanation or hypotheses should be offered. One thought that comes to mind is that by removing the '>= 2 days' constraint might mean radiative influences become more apparent, and result in result in a signal in midsummer.

For both the matched and unmatched cyclone sets the count distribution in August has a significantly different arithmetic mean to that in May at the 95% level according to a two-sided Welch's t-test (without assuming equal variance). However, sequential months are not always significantly different. This information has now been added to the paper.

For comparison the mean monthly numbers of TPVs with Arctic genesis and Arctic cyclones with maximum intensity in the Arctic have been added to Fig. 7. Recall that the constraints for both matched and unmatched cyclones include that the TPVs must have Arctic genesis and that the Arctic cyclones must have maximum intensity in the Arctic. Hence, these constraints have similarly been applied when comparing monthly variability of the matched and unmatched cyclones with that of TPVs and Arctic cyclones. There is little variability in the Arctic cyclone counts. In contrast, the TPV counts have a minimum in July and so follow an evolution similar to that for the matched cyclone counts. This suggests that the number of TPVs limits the number of matched cyclones in midsummer with a consequent maximum in the number of unmatched cyclones then. Crawford and Serreze (2015) show that the strength of the Arctic Frontal Zone is maximum in July (see their Fig. 10) and so this may act to maintain Arctic cyclone numbers in mid-summer despite the reduction in TPVs. The above arguments have now been added to the paper. The July minimum in TPVs could be due to fewer TPVs being generated then because of the seasonal cycle in the tropospheric polar vortex or due to enhanced destruction of TPVs due to tropospheric latent heat release but further research would be necessary to investigate these hypotheses and so we do not include them in the paper.

**Lines 445-447: Role of stratospheric influence in the storm of September 2010 was analysed by Tao W, Zhang J, Zhang X (2017) The role of stratosphere vortex downward intrusion in a long-lasting late-summer Arctic storm. Quart. J. Roy. Meteor. Soc. 143:1953-1966**

Thank you for suggesting this paper. The stratospheric intrusion discussed seems to fit the definition of a TPV in that it exists as a closed vortex in the middle troposphere and lower stratosphere (at 500 and 300 hPa, respectively, see their Fig. 3) when the low-pressure centre at the surface is weak and then "catches" the surface cyclone as both features intensify. However, the authors refer to this feature as the downward intrusion of the stratosphere vortex which, as we discuss in the introduction of our paper, is confusing terminology. In particular, the term "stratospheric vortex" is usually applied to the strong westerly polar stratospheric vortex which exists in winter, but this has not begun to spin up in September (stratospheric winds are easterly during summer with very little disturbance above the influence of the tropopause zone). Consequently, this paper is difficult to cite in relation to the stratospheric influence on Arctic cyclones. However, we have added here a sentence citing this paper as one that presents an Arctic cyclone with equivalent barotropic structure.

Lines 572-575: It would be worthy of note in the paper to mention the importance of this stratospheric influence on rapid development outside the Arctic, e.g., Kouroutzoglou et al., 2015: On the dynamics of a case study of explosive cyclogenesis in the Mediterranean. Meteor. Atmos. Phys., 127, 49-73.

The importance of stratospheric influence on the development of cyclones outside the Arctic (specifically extratropical cyclones) is already discussed in the second paragraph of section 3.4. The paper suggested by the reviewer focuses on cyclogenesis cases during winter over the Mediterranean. We are not specifically considering rapidly developing cyclones and, given the large number of case studies that have been performed on explosively developing cyclones, especially in winter, it does not seem appropriate to pick out an individual study for citation here.

Lines 597-600: A gremlim seems to have gotten into the author list for these two papers by Steve Cavallo and Greg Hakim.

Thank you for pointing this out. The error came from duplicate author names being included in the bibtex file and this has now been fixed.

**Response to reviewer 2:**

**General comments:**

This research investigated the relationship between Arctic cyclone (surface cyclone) and tropopause polar vortices (TPVs) using the same tracking algorithm. The authors analyzed the features of the track density and composited structure for Arctic cyclones by separating matched and unmatched cyclones. The results showed that most of the Arctic cyclones are far from the TPVs at their initiation, and about one-third of the Arctic cyclones developed associated with the TPVs. They also showed that while the genesis of the unmatched cyclones is along the Eurasia coastline, that of the matched cyclone generated over the Arctic Ocean, North America, and the Canadian Arctic Archipelago. The rearward title of relative vorticity for matched cyclones is less than that for unmatched cyclones. The topic is very interesting, and this study shows the relationship between Arctic cyclones and TPVs clearly for the first time. The findings in this study would promote the understanding of the cyclone development over the Arctic in summer. Therefore, the reviewer recommends for publication of this article in WCD after minor revisions.

We thank the reviewer for his/her interest in our research and this positive review.

**Comments:**

Lines 37: Tao et al (2017, QJRMS) also showed the importance of TPVs for the intensification of an Arctic cyclone.

Tao, W., Zhang, J., & Zhang, X. (2017). The role of stratosphere vortex downward intrusion in a longlasting late-summer Arctic storm. Quarterly Journal of the Royal Meteorological Society, 143(705), 1953–1966. https://doi.org/10.1002/qj.3055

Reviewer 1 also suggested inclusion of this paper and it is now cited in the second paragraph of section 3.4. As stated in our response to reviewer 1, the authors of this paper refer to an intrusion of the stratospheric vortex rather than a TPV as having an important role in the development of their case study, although the feature shown seems to fit the definition of a TPV. As the authors do not refer to the feature as a TPV, it does not seem appropriate to include this paper in the list given here of papers that have presented a link between TPVs and Arctic cyclone genesis and intensification.

Line 215: Is the percentage of the cyclone associated with the TPVs within 2° is intermediate between 1° and 3° in Fig. 5? While the author showed the percentage of the cyclone associated with the TPV within 1°, 3°, and 5° at each stage (Fig. 5), the percentage of the matched cyclone (i.e., cyclones satisfied both criteria) is not shown in the results.

The caption of Figure 1 gives the total number of Arctic cyclones and TPVs identified across the 40 extended summer seasons. There is a total of 12155 Arctic cyclones tracked with lifetime of at least 1 day. Of these, 10636 have lifetimes of at least 2 days – this number has now been added to the caption of Fig. 5. The number of matched cyclones (341) is stated in the caption of figure 6. Hence only a small percentage of tracks (3.2%) meet the criteria that they are matched to an Arctic origin TPV because they are within 2° of an Arctic origin TPV at the time of their maximum intensity and within 5° of an Arctic origin TPV at the time of their maximum intensity and prior to their maximum intensity (note this means that their total lifetime will exceed 2 days) and meet their maximum intensity in the Arctic.

We have also added a set of bars to Fig. 5 to show the instantaneous matches within 2 degrees radius at the times of genesis, max growth rate and max intensity (and modified the text accordingly).

**Line 279-281: Do the track density and cyclone features show similar results to the Figs. 1-3 in each month? Or these features have monthly variability?**

We focus on examining the extended summer period as a whole as this improves the robustness of the results by providing more data than if considering the months separately. In response to the reviewer's question though, the genesis densities and track densities are shown for each month in Figs. 1 and 2 below for (all) Arctic cyclones and Arctic genesis TPVs. These panels can be compared to Fig. 1a and 1e in the paper for the extended-summer genesis density of (all) Arctic cyclone and Arctic genesis TPVs, respectively and to Fig. 2a and 2e in the paper for the extended-summer track density of (all) Arctic cyclone and Arctic genesis TPVs, respectively and to Fig. 2a and 2e in the paper for the extended-summer track density of (all) Arctic cyclone and Arctic genesis TPVs, respectively. As expected, the plots are less smooth when considering each of the five months individually compared with the extended-summer period. There is also some month-to-month variability. However, the basic features of the plots, as described in the paper are also present for each month. We have added to the paper that we have also examined the density maps for the individual months, but do not consider it worthwhile to also add Figs. 1 and 2 included here.

---

## Referee Report (RR1)

**General comments:**

This study evaluates how often pre-existing tropopause-level features are present for Arctic cyclone development. Specifically, the the authors seek to quantify the frequency that the tropopause-level disturbance called tropopause polar vortices (TPVs) are linked to Arctic cyclones during several stages of the cyclone's lifecycle. The analysis is performed by computing tracks of TPVs and Arctic cyclones from ERA5 based on certain flow metrics and distance thresholds. Using these methods, the authors find that at most, 10 percent of Arctic cyclones have a nearby TPV at genesis, while 35 and 38 percent of Arctic cyclones are in close proximity to a TPV at maximum growth rate and intensity, respectively. This offers an interesting and somewhat surprising result, because the alternative explanation that is offered is that Pettersen type A cyclogenesis must be more common.

The study is novel in that the relation of TPVs and Arctic cyclones has never before been systematically quantified as has similarly been established in midlatitudes between tropopause-level features and surface cyclone development. I appreciate the authors' thoughtful responses and for incorporating some of my previous points into the manuscript, and think it has improved in the latest version. Some of the major issues, however, have not been adequately addressed, which I describe in detail below. I think consideration of these remaining issues are important in making this manuscript complete and the application of the methodology more strongly convincing.

1) The assumption that Arctic cyclones evolve from a tilted structure to less tilted structure may apply for midlatitudes, but not necessarily the Arctic. While there is no question it applies for baroclinic waves, the Arctic may be a mix of waves, vortices, or both, but should theoretically be dominated by vortices (e.g., Hakim 2000). Unfortunately, Tao et al. (2017) did not specifically consider this aspect in their analysis. Given these differences, and the fact that the authors developed and applied an automatic algorithm to summarize a large sample, it would be highly beneficial if the authors provide a proof of concept example through a case study in their manuscript.

I offer an example of a long-lived TPV and Arctic cyclones in real-time at the writing of this review (31 August) in the following figures. This also would clarify my (or another reader's) possible confusion in applying these methods. I can trace the TPV present just north of Alaska around 77°N 140°W back to at least 9 August using readily available online plots (source provided below), in which the TPV was present in nearly the same location. Thus, it has at least a 23 day lifetime and no clear lysis in sight, making the likely lifetime on the order of at least one month. The previous methodologies by Hakim and Canavan (2005), Cavallo and Hakim (2009), and Szapiro and Cavallo (2018) would likely have all identified this feature as a TPV given its fraction of time spent in the Arctic.

Figure 1 highlights a short 5-day period within this timeframe, in which this single TPV interacted with two Arctic cyclones. Both Arctic cyclones were clearly integrated with the TPV during the later stages of their respective lifecycles. However, it is clear that AC2 would not be associated with the TPV since at its genesis time it was more than 2° away from the TPV (Figure 1c). Both ACs are within 5° of the TPV during around the times of maximum AC growth rates (Figure 1a,b,d). To add to the complexity, it is possible that cyclones similar to AC2 formed from asymmetries on the main TPV that were associated with smaller-scale shortwaves moving around the main TPV and over the Arctic Frontal Zone (See Figure 2 for an example of the subsequent AC forming). This makes me uneasy about how statements, such as the one on lines 357-360, lead the reader to think that the genesis of an Arctic cyclone had nothing to do with a TPV, when it can not be ruled out that it was associated with part

of the mesoscale structure of a large TPV that happened to be centered further away than a 2° (or even a 5°) distance threshold.

This case I highlight is not an outlier case. It could be one TPV and several Arctic cyclones as shown in Figure 1, or several TPVs with one Arctic cyclone as shown in Figure 3. Either way, there is evidence that there are both waves and vortices playing a considerable role, complicating the well-known conceptual picture of baroclinic lifecycles in midlatitudes. I wonder if it is possible that the conclusion that 35 and 38 percent of Arctic cyclones are in close proximity to a TPV at maximum growth rate and intensity, respectively, and/or 10% have a nearby TPV at their genesis time, are misleadingly low due to cases like this. I appreciate that the authors have added some instances in the text pointing out that there is not requirement for sustained associations, but those statements are quite subtle. I would like the authors to address how this case would be analyzed using their current methods, and how or whether it could impact their statistics and conclusions, and to what degree.

With regard to the cross sections composites, I understand that all cyclones are oriented in the same direction relative to their motion (around line 249), and I do not have any issue with whether this impacts the matching. I also understand how the maxima vorticity at each level is projected onto the cyclone motion direction. But I am not clear on how these methods take into account "satellite TPVs" that to break off of a lobe from the main vortex or that happens to be nearby be contributing to the baroclinic growth or maintenance of a single Arctic cyclone. Given that I can find a case currently, it can not be ruled out that this happens quite commonly. Figure 3 highlights an example where one Arctic cyclone intensifies and/or is prolonged by multiple TPVs. In Figure 3a, an Arctic cyclone is intensifying from TPV1. The Arctic cyclone intensifies more rapidly when 2 TPVs are located in close promimity (Figure 3b) until the two TPVs merge (Figure 3c) and TPV2 becomes vertically aligned with the Arctic cyclone (Figure 3d).

In summary, I would like the authors to please, at least, address *from this case*:

(a) which Arctic Cyclone and TPV pairs would be matched,

(b) which Arctic Cyclone and TPV pairs would be unmatched,

(c) how the instantaneous cross sections that are used for making the composites would be oriented. (note that latitude/longitude contours are labeled on the figures)

Note that all analysis plots are available from http://arctic.som.ou.edu/tburg/models/. Archive mode can be found by clicking the settings tab toward the upper right corner.

2) My point about the spectral filtering cutoff at T63 is that the authors should at the very least acknowledge differences from the Cavallo and Hakim (2010) method (i.e., that the TPV identification does not exactly follow their method in this sense). For example, Cavallo and Hakim (2010) use a numerical model with 30-km grid spacing, and as the authors point out in the response, the equivalent spectral resolution is about 60 km in this study. Otherwise in Cavallo and Hakim (2010), it does not appear that there is any filtering, other than the requirement that the TPVs that are selected in the statistical analysis must have a sufficiently large amplitude with respect to the spatial/background pattern. I do not know whether this will cause differences in the sample of TPVs that are identified, however, such differences in the identification method should at the very least be stated to point out that it is possible for this to cause differences. More ideally, however, would be to either use the TPV tracks described Szapiro and Cavallo (2018) or perform a sensitivity test with it, since this would be

more consistent with Cavallo and Hakim (2010). Szapiro and Cavallo (2018) compare their method's climatology with the earlier TPV climatology (not to mention one of the authors is the same as the earlier TPV climatology). To me, this seems like a much stronger justification. The authors pointed out in their response to the review that there are fewer citations for the Szapiro and Cavallo (2018) algorithm than for the TRACK algorithm. This is completely irrelevant and certainly not a reasonable justification. While TRACK has been used a lot, it has been used for other purposes and this is the first time with TPVs. Just because TRACK has been used or cited more in comparison does not imply anything about the consistency of the methods in identifying and/or tracking TPVs as defined in the previous literature.

**Other specific comments:**

1) On line 218, TPVs are further referred to as "Arctic origin" TPVs, which slightly differs from the traditional TPV definition that do not have an origin requirement but do require 60% of their lifetimes in the Arctic are north of 65°N (e.g., Hakim and Canavan 2005; Cavallo and Hakim 2009). If this is indeed correct, then all conclusions that refer to matches between TPVs and Arctic cyclones need to have "Arctic origin" inserted before "TPV" to emphasize the slight distinction (including in the abstract).

2) Lines 71-79: Starting with "Note that TPVS are distinct...", the remainder of the paragraph seems to have no support cited from the literature. While this may be the view of the authors, unless there is documented evidence or evidence is presented, it should not be stated.

3) Lines 80-81: Sentence starting with "Cyclonic TPVs..." is supported in the literature by Cavallo and Hakim (2010).

4) Lines 105-107: Bray et al. (2021) and Lillo et al. (2021) both offer evidence of long-lived cyclonic TPVs in the literature.

5) Lines 541-542: In order to state the statement beginning... "However, this link has been explored...", it needs to be shown that this methodology works as intended. A great way would be through a case study (I highlighted an example above). If the methodology for the intense and long-lived case can clearly demonstrate the intended design, then it would make for a more convincing argument that the algorithm worked for a larger sample that contains less intense or long-lived cases. This is similar to my first major comment above.

**References**

Bray, M., S. M. Cavallo, and H. Bluestein, 2021: Examining the relationship between tropopause polar vortices and severe weather outbreaks. **Accepted and available on early online release**.

Cavallo, S. M. and G. J. Hakim, 2009: Potential vorticity diagnosis of a tropopause polar cyclone. *Mon. Wea. Rev.*, **137 (4)**, 1358–1371.

Cavallo, S. M. and G. J. Hakim, 2010: The composite structure of tropopause polar cyclones from a mesoscale model. *Mon. Wea. Rev.*, **138 (10)**, 3840–3857, doi:10.1175/2010MWR3371.1.

Hakim, G. J., 2000: Climatology of coherent structures on the extratropical tropopause. *Mon. Wea. Rev.*, **128**, 385–406.

Hakim, G. J. and A. K. Canavan, 2005: Observed cyclone-anticyclone tropopause asymmetries. *J. Atmos. Sci.*, **62 (1)**, 231–240.

Lillo, S. P., S. M. Cavallo, D. B. Parsons, and C. Riedel, 2021: The role of a tropopause polar vortex in the generation of the January 2019 extreme arctic outbreak. *J. Atmos. Sci.*, **78 (9)**, 2801–2821, doi:10.1175/JAS-D-20-0285.1.

Szapiro, N. and S. M. Cavallo, 2018: Tpvtrack (v1.0): A watershed segmentation and overlap correspondence method for tracking tropopause polar vortices. *Geoscientific Model Development*, **11 (12)**, 5173–5187.

Tao, W., J. Zhang, Y. Fu, and X. Zhang, 2017: Driving roles of tropospheric and stratospheric thermal anomalies in intensification and persistence of the arctic superstorm in 2012. *Geophys. Res. Lett.*, **44 (19)**.

[Figure]

FIG. 1. 2 PVU tropopause potential temperature (colors) and mean sea level pressure (contours) from GFS analyses valid (a) 18 UTC August 22, (b) 00 UTC August 26, (c) 12 UTC August 26, and (d) 12 UTC August 27, 2021. The TPV in discussion is annotated in gray, while the Arctic Cyclones (ACs) in discussion are annotated in red. I disclose I am not Tomer Burg and I saved these images from his web page and thank him for this useful product available at http://arctic.som.ou.edu/tburg/products/realtime/models/. All data are from the GFS model.

[Figure]

FIG. 2. (a) 2 PVU tropopause potential temperature (colors) and mean sea level pressure (contours), (b) 500 hPa relative vorticity (colors), heights (thick black contours), and 1000:500 hPa thickness (dashed color contours), and (c) omega (colors), 500 hPa heights (black solid contours) and 1000:500 hPa thickness (dashed contours) from GFS 6-h forecasts valid 18 UTC August 27. The TPV in discussion is annotated in gray, while the Arctic Cyclones (ACs) in discussion are annotated in red. A shortwave moving around the TPV is circled in red in panel (b). I disclose I am not Tomer Burg and I saved these images from his web page and thank him for this useful product available at http://arctic.som.ou.edu/tburg/products/realtime/models/. All data are from the GFS model.

[Figure]

FIG. 3. 2 PVU tropopause potential temperature (colors) and mean sea level pressure (contours) from GFS analyses valid 00 UTC (a) August 14, (b) August 15, (c) August 16, and (d) August 17, 2021. The TPV in discussion is annotated in gray, while the Arctic Cyclones (ACs) in discussion are annotated in red. I disclose I am not Tomer Burg and I saved these images from his web page and thank him for this useful product available at http://arctic.som.ou.edu/tburg/products/realtime/models/. All data are from the GFS model.

---

## Author Response (AR2)

**The role of tropopause polar vortices in the intensification of summer Arctic cyclones by Gray, Hodges, Vautrey and Methven: Response to reviewers**

The reviewers' comments are copied below in black with our point-by-point responses in blue. In the paper with edits highlighted, major changes to the paper (mainly larger blocks of text) are highlighted with red font.

**Response to reviewer 1:**

[first paragraph omitted here as identical to previous review]
The study is novel in that the relation of TPVs and Arctic cyclones has never before been systematically quantified as has similarly been established in midlatitudes between tropopause-level features and surface cyclone development. I appreciate the authors' thoughtful responses and for incorporating some of my previous points into the manuscript, and think it has improved in the latest version. Some of the major issues, however, have not been adequately addressed, which I describe in detail below. I think consideration of these remaining issues are important in making this manuscript complete and the application of the methodology more strongly convincing.

We appreciate the reviewer taking the time to provide a second review of our article and respond to their comments below.

1) The assumption that Arctic cyclones evolve from a tilted structure to less tilted structure may apply for midlatitudes, but not necessarily the Arctic. While there is no question it applies for baroclinic waves, the Arctic may be a mix of waves, vortices, or both, but should theoretically be dominated by vortices (e.g., Hakim 2000). Unfortunately, Tao et al. (2017) did not specifically consider this aspect in their analysis. Given these differences, and the fact that the authors developed and applied an automatic algorithm to summarize a large sample, it would be highly beneficial if the authors provide a proof of concept example through a case study in their manuscript.

We do not assume that Arctic cyclones evolve from a tilted structure to a less tilted structure. The structural evolution is a result from the analysis presented in Section 3.4. In the previous revision we added additional text to the Methods section to acknowledge that the TPV features could be waves or vortices. That text has been reordered to reflect the changes to the methods described below and can now be found in lines 251-254.

I offer an example of a long-lived TPV and Arctic cyclones in real-time at the writing of this review (31 August) in the following figures. This also would clarify my (or another reader's) possible confusion in applying these methods. I can trace the TPV present just north of Alaska around 77_N 140_W back to at least 9 August using readily available online plots (source provided below), in which the TPV was present in nearly the same location. Thus, it has at least a 23 day lifetime and no clear lysis in sight, making the likely lifetime on the order of at least one month. The previous methodologies by Hakim and Canavan (2005), Cavallo and Hakim (2009), and Szapiro and Cavallo (2018) would likely have all identified this feature as a TPV given its fraction of time spent in the Arctic.

Figure 1 highlights a short 5-day period within this timeframe, in which this single TPV interacted with two Arctic cyclones. Both Arctic cyclones were clearly integrated with the TPV during the later stages of their respective lifecycles. However, it is clear that AC2 would

not be associated with the TPV since at its genesis time it was more than 2 degrees away from the TPV (Figure 1c). Both ACs are within 5 degrees of the TPV during around the times of maximum AC growth rates (Figure 1a,b,d). To add to the complexity, it is possible that cyclones similar to AC2 formed from asymmetries on the main TPV that were associated with smaller-scale shortwaves moving around the main TPV and over the Arctic Frontal Zone (See Figure 2 for an example of the subsequent AC forming). This makes me uneasy about how statements,
such as the one on lines 357-360, lead the reader to think that the genesis of an Arctic cyclone had nothing to do with a TPV, when it can not be ruled out that it was associated with part of the mesoscale structure of a large TPV that happened to be centered further away than a 2 degree (or even a 5 degree) distance threshold.

The reviewer here states that our method would not associate this particular cyclone (AC2) with the TPV "since at its genesis it was more than 2 degrees away from the TPV". As described in our Methods section (lines 211-224), when determining cyclones that have a **sustained association** with TPVs we match according to the distance between them at the time of maximum growth rate and time of maximum intensity of the cyclone. We do not consider the distance between the TPV and cyclone at the cyclone genesis time. Hence, the reviewer is incorrect to assume our method would be incapable of associating the cyclone and TPV in the case (s)he presents in their review.

The lines referred to by the reviewer describe results for **instantaneous** matches between cyclones and TPVs. In light of changes described below to the radii used for these matches they now state "While TPVs are near to some Arctic cyclones at their genesis time, TPVs are more likely to be near to Arctic cyclones when the Arctic cyclones are intensifying rapidly. For the maximum 10° overlap radius considered, 37% of Arctic cyclones are matched with a TPV at the time of maximum Arctic cyclone growth rate compared to 30% at the time of Arctic cyclone genesis and this difference is proportionally bigger for 2 and 5° overlap radius
Hence, we clearly state the criteria used for our statements. In response to the reviewer's comment though we have added a note here that "(note though that TPVs may influence cyclones from greater distances dependent on their structure (see discussion in Sect.2.2))".

This case I highlight is not an outlier case. It could be one TPV and several Arctic cyclones as shown in Figure 1, or several TPVs with one Arctic cyclone as shown in Figure 3. Either way, there is evidence that there are both waves and vortices playing a considerable role, complicating the well-known conceptual picture of baroclinic lifecycles in midlatitudes. I wonder if it is possible that the conclusion that 35 and 38 percent of Arctic cyclones are in close proximity to a TPV at maximum growth rate and intensity, respectively, and/or 10% have a nearby TPV at their genesis time, are misleadingly low due to cases like this. I appreciate that the authors have added some instances in the text pointing out that there is not requirement for sustained associations, but those statements are quite subtle. I would like the authors to address how this case would be analyzed using their current methods, and how or whether it could impact their statistics and conclusions, and to what degree.

The figures quoted by the reviewer are those in our Fig. 6 (Fig. 5 in previous submission) and are for instantaneous matches between cyclone and TPV features at the relevant times. It is possible to have a cyclone to match with more than one TPV and vice versa. For the sustained matches (when the same cyclone and TPV must match over an extended period), the cyclones are matched to the TPVs so you can get more than one cyclone matching a TPV if the TPV is long enough; this provides the best sample of cyclones. This information has been added to the paper in section 2.2.

With regard to the cross sections composites, I understand that all cyclones are oriented in the same direction relative to their motion (around line 249), and I do not have any issue

with whether this impacts the matching. I also understand how the maxima vorticity at each level is projected onto the cyclone motion direction. But I am not clear on how these methods take into account \satellite TPVs" that to break off of a lobe from the main vortex or that happens to be nearby be contributing to the baroclinic growth or maintenance of a single Arctic cyclone. Given that I can find a case currently, it can not be ruled out that this happens quite commonly. Figure 3 highlights an example where one Arctic cyclone intensifies and/or is prolonged by multiple TPVs. In Figure 3a, an Arctic cyclone is intensifying from TPV1. The Arctic cyclone intensifies more rapidly when 2 TPVs are located in close proximity (Figure 3b) until the two TPVs merge (Figure 3c) and TPV2 becomes vertically aligned with the Arctic cyclone (Figure 3d).

The composites are generated using cross-sections are centred on the cyclones and so will include all upper-tropospheric features (and any TPVs) in the vicinity of those cyclones at the reference time used for compositing.

In summary, I would like the authors to please, at least, address from this case:
(a) which Arctic Cyclone and TPV pairs would be matched,
(b) which Arctic Cyclone and TPV pairs would be unmatched,
(c) how the instantaneous cross sections that are used for making the composites would be oriented. (note that latitude/longitude contours are labeled on the figures)

We thank the reviewer for pointing out this interesting very recent case to us and for using it to highlight his or her questions and concerns about our methodology. We considered applying our approach to this case, but the date is outside the range of data used in the results of the paper and also the ERA5 data were not yet available when we looked on the Copernicus website for the field of potential temperature on the 2 PVU surface which we need to identify the TPV (although the data is now available for vorticity). Hence, it is not possible to look at this case for this paper. To demonstrate our method, we also consider it more appropriate to use a case within the period we have examined (so 1979-2018). In recognition of the reviewer's many questions about our methodology we have made two important changes to the paper:
(a) We have added to the paper a new figure, Fig. 1, to illustrate the application of our criteria for a sample case. We choose a sample case that is already documented in the published literature: one related to the extreme 2016 Arctic cyclone.
(b) We have relaxed our matching criteria so that matched cyclones are now required to be within 10° and 5° of a TPV at the times of maximum growth rate and maximum intensity, respectively (increased from 5° and 2°, respectively). The criteria for unmatched cyclones have been correspondingly changed such that these are greater than 10° from a TPV at both the times of maximum growth rate and maximum intensity. These criteria changes were made to address the reviewer's concerns that our original smaller matching radii may be too restrictive if TPVs and cyclones have structures that are closer to baroclinic waves than point vortices. We have also corrected a minor bug in the matching code used to select cases for the composites which has reduced the numbers of matching cases for given matching radii; consequently, the number of instantaneous matches at the time of maximum growth rate for the new 10° matching radius is similar to that found previously for the 5° matching radius. In practice though the changes in numbers and matching criteria have made little difference to the conclusions drawn from the figures in which the characteristics of the matched and unmatched cyclones are compared as shown in the revised paper.
Finally, in response to point (c) at the end of this reviewer's comment we are confused as, contrary to what is stated by the reviewer, we do not label longitude and latitude on our composite figures (Figs. 9 and 10).

2) My point about the spectral filtering cutoff at T63 is that the authors should at the very

least acknowledge differences from the Cavallo and Hakim (2010) method (i.e., that the TPV identification does not exactly follow their method in this sense). For example, Cavallo and Hakim (2010) use a numerical model with 30-km grid spacing, and as the authors point out in the response, the equivalent spectral resolution is about 60 km in this study. Otherwise in Cavallo and Hakim (2010), it does not appear that there is any filtering, other than the requirement that the TPVs that are selected in the statistical analysis must have a sufficiently large amplitude with respect to the spatial/background pattern. I do not know whether this will cause differences in the sample of TPVs that are identified, however, such differences in the identification method should at the very least be stated to point out that it is possible
for this to cause differences.

We have now explicitly noted that our filtering approach is different to that used by Cavallo and Hakim (2010). Also, now the revised matching radii for the composites are greater relative to the resolution implied by the spectral filtering of the data, so any separation of features greater than the criteria must be very well resolved.

More ideally, however, would be to either use the TPV tracks described Szapiro and Cavallo (2018) or perform a sensitivity test with it, since this would be more consistent with Cavallo and Hakim (2010). Szapiro and Cavallo (2018) compare their method's climatology with the earlier TPV climatology (not to mention one of the authors is the same as the earlier TPV climatology). To me, this seems like a much stronger justification.

The Szapiro and Cavallo (2018) paper published in the journal Geoscientific Model Development introduces a new software package for tracking TPVs. Hence, it is appropriate that the climatology results from the new method are compared there to the previously published climatology of Cavallo and Hakim (2010). We also compare our findings for the climatology to those of Cavallo and Hakim (2010) and other works. Please see section 3.1 of our paper – indeed this first results section was included in our paper largely to demonstrate that our climatology of both TPVs and Arctic cyclones was consistent with that of other studies while also showing, for the first time to our knowledge, a climatology generated using the ERA5 dataset. It is beyond the scope of our study to additionally use the tracks generated using the approach of Szapiro and Cavallo (2018) although, as stated in our earlier response this may be an interesting comparison to perform in a future study.

The authors pointed out in their response to the review that there are fewer citations for the Szapiro and Cavallo (2018) algorithm than for the TRACK algorithm. This is completely irrelevant and certainly not a reasonable justification. While TRACK has been used a lot, it has been used for other purposes and this is the first time with TPVs. Just because TRACK has been used or cited more in comparison does not imply anything about the consistency of the methods in identifying and/or tracking TPVs as defined in the previous literature.

As stated in the Methods section, the identification of TPV feature points is based on the approach used by Cavallo and Hakim (2010). We appreciate that the use of TRACK in many other studies for tracking cyclone features does not automatically make it suitable for tracking the identified TPVs. However, once features have been identified many papers have demonstrated the ability of TRACK to connect the features to form realistic tracks. TRACK has been developed as a general method with a lot of flexibility rather than being designed for a specific application.

Other specific comments:
1) On line 218, TPVs are further referred to as "Arctic origin" TPVs, which slightly differs from the traditional TPV definition that do not have an origin requirement but do require 60% of their lifetimes in the Arctic are north of 65_N (e.g., Hakim and Canavan 2005; Cavallo and

Hakim 2009). If this is indeed correct, then all conclusions that refer to matches between TPVs and Arctic cyclones need to have "Arctic origin" inserted before "TPV" to emphasize the slight distinction (including in the abstract).

It is correct that we require TPVs to have Arctic origin when matching them to Arctic cyclones as described in the Methodology. As can be seen in the bottom row of Fig. 2 (was Fig. 1), relatively few TPVs have genesis outside the Arctic but, as the identified features are not consistent with the generally accepted definition of TPVs (e.g. see the AMS Glossary: https://glossary.ametsoc.org/wiki/Polar_vortex), we wanted to exclude them. The clarifier "Arctic origin" has been added to the Abstract and Conclusions sections.

2) Lines 71-79: Starting with "Note that TPVs are distinct...", the remainder of the paragraph seems to have no support cited from the literature. While this may be the view of the authors, unless there is documented evidence or evidence is presented, it should not be stated.

We have added a citation to a paper by Waugh et al. entitled "What Is the Polar Vortex and How Does It Influence Weather?" to explain the larger-scale stratospheric and tropospheric vortices. It is also discussed in the AMS Glossary (see above).

3) Lines 80-81: Sentence starting with "Cyclonic TPVs..." is supported in the literature by Cavallo and Hakim (2010).

Citation added.

4) Lines 105-107: Bray et al. (2021) and Lillo et al. (2021) both offer evidence of long-lived cyclonic TPVs in the literature.

Thank you for pointing out these very recently published papers. A citation to Lillo et al. (2021) has been added as another example of very long-lived cyclonic TPVs. We have also taken the opportunity to add citations to two other very recently published papers: Capute and Torn (2021) and Lukovich et al. (2021).

5) Lines 541-542: In order to state the statement beginning... "However, this link has been explored...", it needs to be shown that this methodology works as intended. A great way would be through a case study (I highlighted an example above). If the methodology for the intense and long-lived case can clearly demonstrate the intended design, then it would make for a more convincing argument that the algorithm worked for a larger sample that contains less intense or long-lived cases. This is similar to my first major comment above.

See response to first major comment above.

---

## Author Response (AR3)

**The role of tropopause polar vortices in the intensification of summer Arctic cyclones by Gray, Hodges, Vautrey and Methven: Response to editor**

The editor's comments are copied below in black with our point-by-point responses in blue.

- In your response to the reviewer's first comment, you state that "We do not assume that Arctic cyclones evolve from a tilted structure to a less tilted structure.". This is a bit in contrast to your statement in line 240 of the revised manuscript: "These feature separation thresholds were chosen assuming that the Arctic cyclones evolved from a tilted vertical structure to a less tilted vertically-aligned structure at maturity (maximum intensity)", which I think the reviewer was referring to. I understand that the assumption of the tilt is only needed to argue for the smaller required distance at the time of maximum intensity, but maybe you can still adapt the formulation a bit to avoid this contrast between the manuscript and your response.

The reviewer's first comment was "1) The assumption that Arctic cyclones evolve from a tilted structure to less tilted structure may apply for midlatitudes, but not necessarily the Arctic. While there is no question it applies for baroclinic waves, the Arctic may be a mix of waves, vortices, or both, but should theoretically be dominated by vortices (e.g., Hakim 2000)…"

Yes, you are correct of course Stephan that we do impose a smaller distance matching constraint at maximum intensity time than at the maximum growth rate time of the cyclone development. In our response we were referring to the fact that the tilt evolution and composite plots show how the structure of both the matched and unmatched composite cyclones evolve to be approximately barotropic at maximum intensity time – hence the un-tilting is a result of our analysis. Note that the matching distance is rather generous (increased to 5° in the revised paper) at maximum intensity time and so this is unlikely to be a strong constraint, even for tilted systems. The appropriate scales for considering Arctic cyclones as vortices or waves are already discussed in the paper.

- In your response to another comment, you state that: " The composites are generated using cross-sections are centred on the cyclones and so will include all upper-tropospheric features (and any TPVs) in the vicinity of those cyclones at the reference time used for compositing.". I'm not sure that I understand this point: Of course, TPVs in any direction are considered in your matching approach, but in the vertical cross section, I only see anomalies along the direction of cyclone motion, right? So theoretically there may be vorticity anomalies in another direction (e.g., perpendicular to the track) that are not considered in these plots (corresponding to the "satellite TPVs" mentioned by the reviewer). Can you comment on this?

I'm sorry that our response was not clear here. Any TPV that influences the fields along the cross section taken along the direction of the movement of the cyclone will be included in the cross sections. It is possible that there are TPVs that do not strongly affect these fields, though in that case these TPVs would be unlikely to influence the development of the cyclone through the release of baroclinic instability. The following sentence has been added at line 442: "It is assumed that any TPV that affects the evolution of the cyclone lies approximately along these cross-sections, as required for growth by baroclinic instability release." To further demonstrate this point, the figure included here shows the cross-track (S-N if the cyclone travels eastwards) cross section of relative vorticity, potential temperature and potential vorticity through the composite matched cyclone at the time of maximum growth rate. This can be compared to Fig. 9f in the paper, which shows the same plot but for the along-track cross section. As can be seen here, the relative vorticity structure is

approximately vertical, unlike the rearward tilted structure in Fig. 9f. Hence, there is no indication that strong TPV features systematically exist in a direction normal to the cyclone motion at this time.

[Figure]

Figure 1: As for Fig. 9f but in the cross-track direction with the composite cyclone motion directed out of the page.